



# Examining the atmospheric radiative and snow-darkening effects of black carbon and dust across the Rocky Mountains of the United States using WRF-Chem

Stefan Rahimi[1,2], Xiaohong Liu[2,3], Chun Zhao[4,5], Zheng Lu[2,3], and Zachary J. Lebo[2]

**Affiliations**

[1]Institue of the Environment and Sustainability, University of California Los Angeles, Los Angeles, California, 90095, U.S.A.

[2]Department of Atmospheric Science, University of Wyoming, Laramie, Wyoming, 82071, U.S.A.

[3]Department of Atmospheric Sciences, Texas A&M University, College Station, Texas, 77843, U.S.A.

[4]School of Earth and Space Sciences, University of Science and Technology of China, Hefei, China

[5]Anhui Province Key Laboratory of Extreme Events in Atmosphere, University of Science and Technology of China, Hefei, China

*Correspondence to*: Xiaohong Liu (xiaohong.liu@tamu.edu)

**Abstract.** WRF-Chem is run to quantify the in-snow and atmospheric radiative effects of black carbon and dust (BCD, collectively) on a convective-allowing (4-km) grid for water year 2009 across a large area of

the Rocky Mountains. The snow darkening effect (SDE) due to the deposition of BCD on surface snow accelerates the snowmelt by 3 to 12 millimeters during late spring and early summer, effectuating runoff increases (decreases) prior to (after) June. Meanwhile, aerosol radiation interactions (ARI) associated with BCD generally dim the surface from incoming solar energy, introducing an energy deficit at the surface and lead to snowpack preservation by 1 to 5 millimeters. Runoff alterations brought forth by BCD ARI are of

opposite phase to those associated with BCD SDEs, and the BC SDE drives a majority of the surface energy and hydrological perturbations. More generally, changes in snow water equivalent (SWE) brought forth by BCD effects are due to perturbations to the surface energy budget and not initiated by changes in precipitation amount or type. It is also found that perturbations to the surface energy budget by dust ARI can differ in sign from those of BC ARI, with the former being positive across high-albedo surfaces,

enhancing snow melting and changing runoff.



## 1. Introduction

The arid Rocky Mountains of the western U.S. (WUS) receive most of their precipitation in the form of snowfall from October through March. The resulting snowpack forms a network of natural

mesoscale storage reservoirs that provide approximately 80% of the surface water across the region during the warm season (Serreze et al., 1999; Hamlet et al., 2007). All life in the region fundamentally depends on the timed release of runoff from snowpack; humans rely on these resources for agriculture and power generation. In recent decades however, there have been observed changes in the hydrology across the WUS associated with external climate forcings (e.g., anthropogenic climate change) that may be acting to

compromise the security of water resources across the region and beyond (Hamlet et al., 2007).

Numerous studies have shown that annual maximum snow water equivalent (SWE) values have decreased since 1950 (Das et al., 2009; Mote 2006; Pierce et al., 2008), increasing (decreasing) runoff discharge in the winter and spring (summer) (Rajagopalan et al., 2009; Qian et al., 2009). Externally forced warming associated with greenhouse gases and light-absorbing aerosols (LAA; Pierce et al., 2008) and

LAA deposition on snowpack (Flanner et al., 2007; Qian et al., 2009, Wu et al., 2018), rather than natural climate variability, are believed to be the major contributors to this decrease.

LAA such as black carbon (BC) and dust (refereed to collectively as BCD) can affect the hydrology across the WUS as they interact with sunlight, altering the vertical thermodynamic structure of the atmosphere. These aerosol-radiation interactions (ARI) may lead to changes in precipitation amount

and type (Pederson et al., 2011). BCD may also deposit on snowpack, increasing the surface absorptivity and enhancing melting in a process referred to as the snow darkening effect (SDE; Warren and Wiscombe, 1980; Painter et al., 2007). Surface warming is generally brought about by the SDE, while the surface can either cool or warm from ARI. Both effects have been shown to be important across the region however, especially since the surface radiative budget is sensitive to small perturbations in albedo (Pepin et al.,

55   2015).

BCD aerosols find their way into the WUS from both near-field and far-field sources. BC, produced via the incomplete combustion of fossil fuels and biomass burning, is primarily emitted in WUS cities and by wildfires (Bond et al., 2013). Dust meanwhile is primarily emitted from southwestern U.S. deserts via wind erosion (Tegen et al., 2004). Following emission, both aerosols are transported


downstream by prevailing westerlies toward the Rocky Mountain region. Additionally, a sizable

component of atmospheric dust across the WUS originates from Asian sources (Fischer et al., 2009).

BCD SDEs and ARI across the WUS have been studied using global climate models (GCMs)

(Flanner et al., 2007; Qian et al., 2009; Wu C. et al., 2018) and regional climate models (RCMs) (Wu L. et

al., 2018), each with their own advantages and drawbacks. Heterogeneous mesoscale meteorology features

(i.e. precipitation, temperature, and snow characteristics) can be simulated better with RCMs than GCMs,

as higher grid resolutions are typically used. Wet removal by precipitation rather than dry removal is a

more effective pathway for cleansing the atmosphere of BCD (Zhao et al., 2014). Therefore, high-

resolution (cloud-resolving) simulations, through their more physically based treatment of orographically

forced precipitation, should better simulate these aerosols' lifecycle. Additionally, RCMs such as the

Weather Research and Forecasting model coupled with Chemistry (WRF-Chem; Grell et al. (2005)) have

chemical and aerosol options that are generally more sophisticated than typical GCMs.

While WRF-Chem has been used to study the impacts of BCD SDEs and ARI across California on

convective-permitting scales (Wu L. et al., 2018), the application of this model to the American Rockies as

a whole has not been made. Smaller inner-continental cities and municipalities across this zone may be

highly sensitive to changes in hydrology brought about by BCD effects, hence providing motivation for this

study.

Using WRF-Chem, we seek to quantify perturbations to WUS meteorology and hydrology

induced by BCD for water year 2009. Sensitivity experiments are run on convective-allowing scales (4 km

grid spacing) to isolate the effects of BCD SDEs and ARI on temperature, precipitation, snow, and runoff.

This study begins with a presentation of the model, methodology, and data in section 2. Section 3

provides a meteorological and chemical validation of WRF-Chem. The radiative effects of BCD associated

with SDEs and ARI are explored in section 4, and their effects on WUS weather are examined in section 5.

Section 6 briefly evaluates the implications of undersimulated dust emissions, while Section 7 examines the

combined effects (SDE+ARI) of BCD. Conclusions are presented in Section 8.


## 2. Model, Experiments, and Data

### 2.1 WRF-Chem



In this study, WRF-Chem 3.5.1 updated by University of Science and Technology of China (USTC) is used. This USTC version of WRF-Chem includes capabilities not available in publicly released

WRF-Chem versions such as the diagnosis of radiative effects of aerosol species, land surface coupled biogenic VOC emissions, and aerosol-snow interactions (Zhao et al., 2013a,b, 2014, 2016; Hu et al., 2019).) The model is run on a 4-km grid with 50 vertical levels across a large portion of the WUS (Figure 1). The SNow, ICe, and Aerosol Radiative (SNICAR) model (Flanner et al., 2007), which uses BCD deposition flux from the atmosphere to treat the SDE and treats ice and snowpack aging, was coupled into

the Community Land Model version 4.0 (CLM4; Oleson et al., 2010 ) by Zhao et al. (2014). SNICAR uses snow water (both ice and liquid) and aerosol loading information to compute the snowpack's radiative properties within multiple snow layers based on the theories by Warren and Wiscombe (1980) and Toon et al. (1989). The utility of SNICAR in simulating the albedo reductions in snow has been tested in laboratory experiments (Hadley and Kirchstetter, 2012).

The MOdel for Simulating Aerosol Interactions and Chemistry (MOSAIC; Zaveri et al., 2008) and the Carbon Bond Mechanism version "Z" (CBM-Z; Zaveri and Peters, 1999) photochemical model are used to treat aerosol and atmospheric chemistry. MOSAIC uses a 4-bin sectional approach to simulate the aerosol size distributions of BC, dust, sulfate, ammonium, nitrate, organic matter, and sea salt for radii of 0.039-10 $\mu$m. Additionally, MOSAIC treats the processes of aerosol nucleation, coagulation, condensation,

water uptake, and aqueous chemistry. Aerosol dry deposition is handled via the method in Binkowski and Shankar (1995), which includes Brownian and turbulent diffusion as well as gravitational settling. Wet deposition of aerosols and gases by in-cloud and below-cloud precipitation scavenging is treated following Easter et al. (2004). Similar to Zhao et al. (2013a) and Wu et al. (2017; 2018), aerosols are assumed to be internally mixed within each size bin. Aerosol optical properties such as extinction, single-scatter albedo,

and asymmetry parameter are computed as a function of wavelength at each grid point based on the bin- and volume-averaged refractive index for each aerosol species (Fast et al. 2006). ARI are treated in the radiation code via the methodologies in Zhao et al. (2011), in which the direct radiative effects is computed diagnostically using the method in Ghan et al. (2012). Aerosol radiative feedbacks are enabled, and the number concentration of activated aerosols is treated prognostically in the cloud microphysics scheme

(Morrison et al. 2009) following Gustafson et al. (2006).



### 2.2 Emission data

Anthropogenic emissions from the Environmental Protection Agency's (EPA) 2011 National

Emission's Inventory (EPA NEI-11; https://www.epa.gov/air-emissions-inventories/2011-national-

emissions-inventory-nei-data) are used. These emissions contain location-specific point and area source

emissions and are interpolated to a 4-km grid using the open-source software emiss_v04.F

(ftp://aftp.fsl.noaa.gov). Biomass burning emissions, available on a ~1 km grid from the Fire INventory

from the National Center for Atmospheric Research (NCAR) (FINN; Wiedinmyer et al., 2011), are used;

FINN makes use of satellite and land coverage observations to estimate emissions from wildfires. Both

EPA NEI-11 and FINN data are updated hourly to account for the diurnal cycle of their respective

emissions. Biogenic emissions of isoprene and monoterpenes are calculated online following Guenther et

al. (1993). In-domain dust emissions are also calculated online following Zhao et al. (2010, 2013b) using

the Goddard Global Ozone Chemistry Aerosol Radiation and Transport (GOCART) dust scheme from

Ginoux et al. (2001). Short test simulations showed that surface dust concentrations were underpredicted.

Therefore, the dust-tuning factor was increased from 1.0 to 1.2 in the six WRF-Chem experiments.

### 2.3 Chemistry boundary and initial data

### 2.3.1 Non-dust aerosols

Most far-field cross boundary and initial condition chemistry for WRF-Chem is handled using the

open-source software mozbc (NCAR, accessed 2018 at https://www.acom.ucar.edu/wrf-

chem/download.shtml). This software uses simulated chemical output from the Goddard Earth Observing

System version 5 (GEOS-5) model to generate the chemical initialization and lateral boundary condition

files for WRF-Chem based on MOZART-4 (Model for OZone And Related Tracers version 4) simulations.

Chemical boundary tendencies are updated every 6 hours. Relevant to this study, mozbc is used to provide

the lateral boundary and initial condition chemistry for many chemical constituents and aerosols, with the

exception of dust (see Section 2.3.2).

### 2.3.2 Dust



In this study, to avoid the issue of different dust bin cutoff sizes between MOZART-4 and

MOSAIC, we ran a quasi-global (QG) WRF-Chem simulation to provide initial and cross-boundary dust in

our simulations. The QG WRF-Chem was ran at 1° horizontal grid spacing with the identical options to the

convective-permitting experiments except that convection was parameterized following Kain (2004). The

QG experiment was ran on a 360×130 grid (180°W-180°E, 60°S-70°N) following the methodology of Zhao

et al. (2013b) and Hu et al. (2016). The QG experiment was initialized on 2 December 2008 00:00 UTC

and integrated until 1 August 2009 00:00 UTC. Horizontal winds and temperature were nudged in the QG

experiment to meteorological fields from the Climate Forecast System Reanalysis (CFSR; Saha et al.,

2010) with a relaxation coefficient of 0.003. A dust tuning factor of 0.6 was needed in the QG experiment

to appropriately simulate spatially averaged dust optical depth values. Dust from the QG experiment was

interpolated to grid cell locations at the lateral boundaries of the convective- permitting experiments via a

third-order least squares regression method. More details about the QG WRF-Chem simulation can be

found in Zhao et al. (2013b) and Hu et al. (2016).

**2.4 Experimental setup**

Meteorological forcing at initialization and at the lateral boundaries of the convective-permitting

experiments is provided from CFSR. Sea surface temperatures in the QG experiment and atmospheric

tendencies in the QG and convective-allowing experiments are updated every 6 hours. Convection is not

parameterized in the 4-km experiments. A detailed listing of the physical parameterizations and physics

packages used in the 4-km experiments are shown in Table 1.

The original control experiment was initialized on 26 September 2008 at 00:00 UTC and run

through 1 August 2009. However, this experiment was found to underpredict snow water equivalent (SWE)

by several hundred millimeters due to underpredicted precipitation across the Rocky Mountains (not

shown). A companion WRF simulation run without chemistry (NOCHEM) was found to significantly

outperform WRF-Chem in simulating Rocky Mountain snowpack when compared to ground-based

measurements; hence, a new set of WRF-Chem experiments was designed.

We restart our WRF-Chem simulations on 1 February 2009 00:00 UTC using surface energy and

hydrological fields from the NOCHEM restart file. Six "branch" WRF-Chem simulations are launched





from the new restart file to quantify the impacts of BCD across the Rocky Mountains (see Table 2). These experiments, to be run through 1 August 2009, consist of the following:

1.      CNT – the control experiment simulates both the SDEs and ARI associated with BCD. CNT also includes the indirect effects associated with various aerosols, as the number concentration of activated aerosols is calculated based on the local aerosol characteristics in each grid cell.

     2.      noSDE – this simulation is identical to CNT except that the deposition fluxes of BCD and snow BCD loading are set to zero in CLM/SNICAR. The deposition fluxes in the atmospheric

component of WRF-Chem remain unchanged, effectively allowing BCD to vanish as they are removed from the atmosphere. The ARI associated BCD remain in this perturbation experiment.

     3.      noARI – this simulation is identical to CNT except that the mass contribution of BCD to the total aerosol optical properties is set to zero in the radiation code. Specifically, the contributions

of BC, dust, and calcium to the atmospheric radiative effects are removed.

     4.      noBCD – this simulation is identical to CNT except that the SDEs and ARI due to BCD are removed.

     5.      noBCSDE – this simulation is identical to noSDE except that only the BC SDE is removed.

     6.      noBCARI – this simulation is identical to noARI except that only BC ARI are removed.

By examining the differences between the results of the six simulations, species-specific SDEs and ARI associated with BCD can be quantified across 4 subregions of the Rocky Mountains shown in Figure 1. These subregions include (i) Greater Idaho, (ii) the Northern Rockies, (iii) the Utah Mountains, and (iv) the Southern Rockies; we consider elevations equal to or greater than 2,200, 2,400, 2,200, and 2,600 meters, respectively, across the 4 subregions. These regions were chosen because the water resources

of these four areas depend heavily on the timing of local snowcover melt and orographic precipitation event characteristics. These watershed-scale numerical estimates can serve to inform local policymakers, planners, and industries on the impacts of BCD effects. While BC SDEs and BC ARI are explicitly quantified by the difference between CNT and noBCSDE (noBCARI), it must be born in mind that dust SDEs (ARI) are taken to be the linear difference between noSDE and noBCSDE (noARI and noBCARI).




### 2.5 Observational data

The performance of CNT and NOCHEM in simulating several important meteorological variables is first evaluated in this study. Daily point-source measurements of minimum ($T_{min}$), maximum ($T_{max}$), and average ($T_{av}$) 2-m temperature, as well as precipitation and SWE from 418 SNOw TELemetry (SNOTEL;

Serreze et al., 1999) sites across the WUS are used to evaluate the model performance (see black dots in Figure 1). The spatial distribution of simulated monthly $T_{min}$, $T_{max}$, $T_{av}$, and precipitation are evaluated using the Precipitation-elevation Regressions on Independent Slopes Model (PRISM; Daly et al., 1994). PRISM fields are available on a ~4-km mesh. The spatial variability of monthly SWE is also evaluated in this study against data from the University of Arizona (UA). These data are generated with the

methodology used to generate PRISM and are mapped to a ~4-km grid (Broxton et al., 2016). Model simulated snow cover fraction (SCF) is evaluated against the High-resolution (0.05º) measurements from the MODerate resolution Imaging Spectroradiometer (MODIS) Aqua (Hall and Riggs, 2016).

Simulated BCD are compared to measurements from 23 Interagency Monitoring of PROtected Visual Environments (IMPROVE; Malm et al., 1994) network sites (see yellow triangles in Figure 1).

Simulated BC concentrations are approximated to be fine-mode elemental carbon (EC) from IMPROVE. Although EC can be quite different from BC, this type of comparison has been made in other studies (e.g., Liu et al., 2012; Wu et al., 2018). Simulated dust concentrations are approximated using the methods in Kevouras et al. (2007) by adding fine-mode soil to the difference between particulate matter (PM) having a dry-size diameter of less than 10 $\mu$m (PM$_{10}$) and PM having a dry-size diameter of less than 2.5 $\mu$m

(PM$_{2.5}$). This approximation was found to be reliable at 9 inland rural IMPROVE sites in Malm et al. (2007), at which the dust contribution to the coarse mode was 74-90%.

### 3. Evaluation of simulated meteorology, BC, and dust

### 3.1 Meteorological variables

Time series of simulated and SNOTEL 2-m temperature, precipitation, and SWE from 1 October 2008 and onward are shown in Figure 2. NOCHEM exhibits a $T_{av}$ cold bias of 2.41ºC compared to the 418-site SNOTEL average of 1.60ºC. Precipitation and SWE are overestimated by NOCHEM, which simulates



a bias of + 0.97 mm d⁻¹ and +9.61 mm, respectively. Despite these biases, NOCHEM-simulated $T_{av}$,

precipitation, and SWE are highly correlated with SNOTEL, with Pearson $r$-values of +0.97, +0.83, and

+0.97, respectively.

Evaluation of CNT is performed from 1 February 2009 onwards. CNT and NOCHEM simulate

lower $T_{av}$ values than the 418 SNOTEL-site average; -1.67°C and -1.84°C, respectively. CNT is warmer

than NOCHEM, although both simulations have $r$-values of +0.98 compared to SNOTEL. CNT and

NOCHEM are characterized by wet bias of 0.71 mm d⁻¹ and 1.18 mm d⁻¹, respectively, and both simulations

have $r$-values of +0.81. Despite having $r$-values of +0.98, CNT and NOCHEM overpredict SWE by 5.6

mm and 16.6 mm, respectively. The reduced precipitation bias in CNT may be the related to the prognostic

number concentration of activated aerosols, but this result is not verified in this study.

The spatial distributions of February-through-July-averaged $T_{av}$, $T_{min}$, and $T_{max}$ for CNT and

PRISM are shown in Figure 3, where large terrain-induced heterogeneity in temperature can be seen. For

validation purposes, we exclude simulation results near CNT's lateral boundaries in an attempt to remove

gridpoints whose solutions are relaxed to coarse-resolution CFSR. All subsequent statistics make use of

CNT data only within the black box shown in Figure 3 (37°N to 47°N; -116.5°E to -103.5°E). The 2-m

temperature pattern is similar between CNT and PRISM as indicated by $r$-values of +0.94, +0.85, and

+0.96 for $T_{av}$, $T_{min}$, and $T_{max}$, respectively. However, a warm bias in $T_{av}$ (0.69°C), driven primarily by a

warm bias in $T_{min}$ (2.00°C) is simulated by CNT.  Meanwhile, cold biases in $T_{av}$ and $T_{min}$ are simulated at

high elevations. The overall temperature bias is negatively correlated with elevation; the bias becomes

more negative with increasing elevation (third column of Figure 3). Negative $r$-values between simulated

bias and gridcell elevation ($r_{ZB}$) in excess of 0.68 are computed for $T_{av}$, $T_{min}$, and $T_{max}$. Local cold (warm)

biases across mountain chains (basins) may be related to CNT's overprediction (underprediction) of SWE

and SCF within these zones (Figure 4).

The spatial distributions of February-through-July-averaged precipitation rate, SWE, and SCF are

shown in Figure 4. These hydrological fields do not correlate as well with observations as in the case of 2-

m temperature, with $r$-values for CNT/PRISM precipitation rate, CNT/UA SWE, and CNT/MODIS SCF of

+0.78, +0.90, and +0.82, respectively. Domain-averaged precipitation rate is overpredicted in CNT by 0.59

mm d⁻¹, with biases larger than 1 mm d⁻¹ simulated locally at higher elevations. This wet bias is also





evident in SWE and SCF comparisons, where CNT overpredicts SWE by more than 75 mm and SCF by

30% in several mountain chains. CNT simulates a mean wet bias that increases with increasing elevation;

$r_{ZB}$-values of +0.37, +0.20, and +0.38 are recorded for precipitation rate, SWE, and SCF, respectively.

**3.2 BC and dust**

The spatial distributions of in-domain February-through-July-averaged anthropogenic BC and dust

surface emission rates are shown in Figures 5a and 5b, respectively. BC emissions are maximized within

cities of the WUS such as Denver, Salt Lake City, and Las Vegas. Anthropogenic BC emissions also

collocate with the federal interstate network. We note that, especially during summer months, a sizeable

fraction of the BC emissions is associated with wildfires (not shown). Wildfires are scattered during our

simulation record (not shown), although they emit 100s to 1000s of milligrams per $m^2$ of BC into the

atmosphere. Dust emissions on the other hand are maximized over the Mojave and Great Basin deserts as

well as eastern Montana. As shown in Figure 5c and 5d, the largest burdens of BCD are simulated near or

atop their respective emission sources. However, CNT emits over an order of magnitude more dust than

BC, contributing to a large difference in February-through-July-averaged burdens (0.31 mg m$^{-2}$ for BC, 9.9

mg m$^{-2}$ for dust).

Time series of BC and dust surface concentrations ([BC] and [dust], respectively) from CNT and

IMPROVE are shown in Figures 6a and 6b, respectively. CNT slightly oversimulates [BC] from February

through April, and undersimulates [BC] from June through July, but otherwise compares well with the site-

averaged [BC] from IMPROVE. CNT and IMPROVE agree on site-averaged [BC] between 0.05-0.11 $\mu$g

m$^{-3}$, with CNT slightly underpredicting [BC] by ~0.01 $\mu$g m$^{-3}$. Despite this agreement between CNT and

IMPROVE, CNT simulates a modest temporally averaged [BC] spatial correlation (*r*-value) of +0.37

(Figure 6c).

Dust is undersimulated by CNT throughout the period of simulation (Figure 6b and 6c). CNT has

a February-through-July bias of -2.73 $\mu$g m$^{-3}$ with the IMPROVE mean of 4.38 $\mu$g m$^{-3}$. CNT [dust]

correlates better spatially with IMPROVE than does [BC], with a *r*-value of +0.61. Simulated dust

emission sources are prevalent within our domain. However, our domain could not explicitly capture major

dust emission sources across the southwestern U.S., nor does the Ginoux et al. (2001) scheme capture dust





emissions from roadways. In light of the simulated low bias in CNT, the dust tuning factor (DTF) was

increased to 2.0 in a seventh experiment, DTF=2 (green line in Fig. 6b). However, despite a 60% increase

in dust emissions (1.2/2.0 = 0.6), DTF=2 still simulates a February-through-July low bias in [dust] of 1.84

$\mu$g m$^{-3}$ (~43% lower than IMPROVE). The increase in the DTF was arbitrary. Due to the high

computational demand of WRF-Chem experiments, further simulations with increased DTFs values were

not conducted. The differences between the DTF=2 experiment and CNT will be discussed later.

The undersimulation of [dust] within our domain may be due to many factors. In the QG WRF-

Chem experiment, dust AOD was chosen as the primary reference metric for tuning purposes (see Rahimi

et al. 2019), not [dust] or dust burden. Specifically, the QG WRF-Chem (Community Earth System Model;

CESM) simulation integrated a quasi-globally (65ºS-65ºN) February-through-July averaged AOD of 0.025

(0.035 from CESM). The smaller dust AOD in the QG experiment relative to the reasonable GCM value

may be tied directly to underpredicted [dust] across the WRF-Chem domain, as cross-boundary dust

transport was probably undersimulated. Biases in [dust] may also be attributed to regional underestimations

in dust emissions from roadways as well as small-scale meteorological dust sinks within our domain. The

vertical distribution of dust may have been misrepresented in our simulations, too.

**4. Radiative effects of BC and dust**

    Radiative effects of BCD are computed diagnostically following Ghan et al. (2012). For

computation of the atmospheric radiative effects at the TOA and the surface, the radiation scheme is run

interactively with all aerosol species factored into the calculation of beam transmittance (e.g. upwell

shortwave flux at the TOA, downwelled longwave flux at the surface, etc.). The radiation scheme is then

called iteratively in a methodology that sees, for example, BC excluded from the calculation of beam

transmittance variables. By subtracting the transmittance variables in the case where BC is excluded from

the case including all aerosol species, the radiative effect of BC can be quantified. The same procedure is

applied to quantify dust transmittance.

**4.1 Atmospheric radiative effect**





Figure 7 shows that BC and dust have opposite clear-sky TOA radiative effects across the CNT domain. February-through-July domain-averaged radiative effects of +0.55 and -0.98 W m$^{-2}$ (-1.09 W m$^{-2}$ in DTF=2) are simulated for BC and dust, respectively. The largest positive BC radiative effect (RE) at the TOA occurs over BC emission source regions (cities and power plants; Figure 7a), while the most negative

dust-induced RE at the TOA occurs over deserts (Figure 7b). Note that Figure 7b depicts the negative of the dust TOA RE. The spatial distribution of the BCD-induced REs correlate with the spatial distribution of these aerosols' respective burdens (Figure 5). The BC-induced RE tends to be less positive over mountainous areas, while dust-induced cooling tends to be less negative. Of the four subregions, the Utah Mountains are characterized by the most positive TOA RE of +0.59 W m$^{-2}$ by BC (Figure 7a) and the most

negative TOA RE of -0.94 W m$^{-2}$ by dust (Figure 7b; -1.06 W m$^{-2}$ in DTF=2).

Table 3 breaks up the BCD TOA radiative effects by month. The BC TOA RE is positive across all subregions; the effect is largest across the Utah Mountains (+0.81 W m$^{-2}$) in April, as these mountains sit directly east of Salt Lake City and adjacent anthropogenic BC emission sources. For dust, the TOA radiative effect is largest across the Utah Mountains compared to the other subregions. Here, the TOA RE

reaches a base in May of -1.11 across the Utah Mountains. It is noteworthy that the Sothern Rockies see a May TOA RE of -1.05 W m$^{-2}$ , but this region's TOA RE is comparable to other subregions during other months. The more negative TOA radiative effect due to dust across the Utah Mountains is due to the fact that this area sits immediately downstream of dust source regions.

**4.2 In-snow radiative effect**

Figures 7c and 7d show that the in-snow radiative effect (ISRE) varies substantially with topography (see Figure 1) when considering the February-through-July mean. BC dominates perturbations to the surface energy budget compared to dust over snow-covered areas with ISRE values in excess of 2 W m$^{-2}$ integrated over mountainous terrain. Dust-induced ISRE values increase with southward extent,

maximized locally over the Utah Mountains and Southern Rockies. The ISRE values and patterns simulated here are consistent with those of Wu et al. (2018) who used a variable-resolution version of CESM to compare the SDEs of BCD.  It is remarkable that BC ISREs dominate over dust considering that top snow




layer burdens of dust are ~2.5 orders of magnitude larger than those of BC (7.6 mg m$^{-2}$ for dust compared to 0.0178 mg $^{-2}$ for BC).

340        Table 4 reinforces the finding that BC contributes to a stronger ISRE than dust across our four subregions in CNT. This result is unsurprising considering WRF-Chem experiments underpredict [dust] and reinforces the finding by Wu et al. (2018), who simulated a significantly larger BC ISRE compared to that of dust across the region (Wu et al. 2018). BC ISRE are maximized in May across Greater Idaho and the Northern Rockies, with values of +0.73 and +1.09 W m$^{-2}$, respectively. Dust ISRE values across these

regions are about a quarter of the magnitude comparatively, even in the DTF=2 experiments (dust-induced ISREs are larger compared to CNT). Across the Utah Mountains and the Southern Rockies, the BCD ISRE peaks in April and May, respectively. The peak in ISRE during April across the Utah Mountains, occurring one month earlier than the other subregions, is due to maximized upstream dust emissions during April (not shown), which drives a fractionally larger dust-induced ISRE across this region.


## 5. BCD SDE and ARI impacts on WUS weather and hydrology

### 5.1 SDE

#### 5.1.1 Spatial patterns of SDE anomalies

        From March through June, BCD SDE brings forth relatively small 2-m temperature increases of

0.05ºC to 0.5ºC across portions of Greater Idaho, the Northern Rockies, the Utah Mountains, and the Southern Rockies (Figure 8a) compared to Wu et al. (2018). This warming is comparable however to SDE-induced warming reported in Qian et a. (2009). Similar to Wu et al. (2018) and Qian et al. (2009) is the fact that the strongest simulated warming does not generally occur across the highest terrain. In fact, the largest SDE-induced warming occurs across the western upslope regions of the Northern Rockies, the Utah

Mountains and the Southern Rockies. Here, we defined the western upslope areas of our subregions to be western portion of our most highly elevated terrain where the terrain height increases with eastward extend. BC contributes to most of the SDE-induced warming across the northern subregions (Figure 8b) compared to that of dust (Figure 8c), while BC- and dust-induced warming of the air temperature are more similar across the southern subregions. This aerosol-warming pattern is generally correlated with the BCD ISRE





pattern in Figure 7, where BC ISREs far outweigh those of dust across the northern subregions but are more

comparable across the southern subregions.

The ISREs of BCD are highly correlated with general reductions in SWE across our four

subregions (Figure 8d). Driven primarily by in-snow BC (Figure 8e), SWE reductions of between 2.5 and

10 mm are simulated. Across the Utah Mountains and Southern Rockies, our simulated SWE reduction

pattern is consistent with Wu et al. (2018) while our anomaly magnitudes are smaller; Wu et al. (2018)

simulated SWE reductions between 10-50 mm across these mountains during springtime (March-through-

May). We present the March-through-June average, hence our simulated anomalies are different than Wu et

al. (2018) comparatively. Another factor potentially contributing to SWE anomaly differences is that Wu et

al. (2018) used a GCM that explicitly treated large-scale feedbacks whereas our study does not. These

large-scale feedbacks led to an increase in SWE across Greater Idaho and the Northern Rockies in Wu et al.

(2018), a feature not captured in this study. It should be noted that the BCD effects on WUS meteorology

and hydrology considered here are purely local; they are not due to BCD SDEs and ARI beyond the

domain boundaries. More generally, Greater Idaho, the Northern Rockies, and the Utah Mountains see

simulated SWE reductions across most elevated surfaces above 2,200 m. Across the Southern Rockies

meanwhile, reductions in SWE are mostly confined to the western portion of the mountainous terrain.

The reductions in SWE are driven by snow impurities that reduce snowpack albedo by as much as

0.02 across many elevated areas (Figure 8g). These albedo reductions are driven mainly by BC (Figure 8h).

As the snowpack absorbs more heat, melting rates within the top snow are enhanced (not shown), leading

to ice crystal growth of the underlying snow at the expense of liquid (Hadley and Kirchstetter, 2012).

Figure 8j shows that snow grain radii are mostly enhanced by several microns across snow-covered regions

from March through June.

The geographic distribution of BCD SDE-induced anomalies can be tied to the mean normalized

BCD burdens in the top snow layer (Figure 9). Burdens of BCD in the top snow represented in Figure 9 are

divided by their respective means and are therefore unitless. The normalized in-snow burdens of BC are

generally largest across western upslope regions of our subregions (Figure 9a). Different from in-snow BC

burdens, in-snow dust burdens generally decrease with northward extent across our domain (Figure 9b).

This is due to the fact that southern mountain ranges sit directly downstream of dust emission sources,



while northern mountain chains do not. Additionally, the largest in-snow BCD mass mixing ratios are generally found across western upslope regimes where 1) aerosol deposition fluxes are maximized (not

shown) and 2) snow burdens are relatively smaller to those further uphill (see Figure 4, middle row) leading to maximized in-snow aerosol mass mixing ratios. The areas of maximum warming tend to be located within these moderately elevated upslope zones, while the largest SDE-induced reductions in SWE tend to occur at higher elevations downwind, where SWE is largest. The SWE results differ from those of Wu et al. (2018), who found that the maximum reductions in SWE occurred across many basins of the

WUS, especially across the northern subregions, although our result agrees with those of Qian et al. (2009). It should be noted that WRF-Chem experiments undersimulate SWE and SCF across low- and moderately-elevated regions such as the Snake River Basin (southern Idaho) and the Green River Basin (southwestern Wyoming). Simulated snow coverage may therefore have been too low for consequential SDE-induced perturbations to the meteorology across these areas, leading to the discrepancies between this study and Wu

et al. (2018).

### 5.1.2 Timing of SDE

From Figure 10a it can be seen that BCD SDEs induce regionally averaged warming by no greater than 0.2°C across our four subregions. The largest simulated warming occurs across the Utah Mountains in

late April and Greater Idaho in mid-June (0.15°C). BC almost universally heats the surface air (crosses, Figure 10a), but we do note several instances where the dust SDE cools the surface air (hollow circles, Figure 10a). This is most probably the result of the assumption of linearity made in quantifying dust SDE as the difference between noBCSDE and noSDE (Section 2.3.2).

Peak SWE reductions are relatively well correlated with peak 2-m temperature increases and

maximal ISRE values across the southern subregions. Peak SWE reductions of between 8 and 10 mm occur in mid-April and mid-May across the Utah Mountains and Southern Rockies (between 4 and 5%), respectively. These reduction maxima, driven mainly by BC SDEs, occur concurrently with seasonally maximized ISRE values of +4 to +5 W m⁻² (Figure 11). It is noteworthy that SWE reductions across these southern subregions have a larger dust SDE-induced component than simulated across northern subregions.

As a percentage, negative SDE-induced SWE anomalies enlarge as the warm season progresses, with SWE





simulated reductions of ~65%, ~10%, ~7%, and ~50% occurring across Greater Idaho, the Northern Rockies, the Utah Mountains, and the Southern Rockies, respectively by mid-July. The progressive increase in SWE reduction percentages throughout the warm season is the result of decreased overall snowpack concurrent with increasing in-snow BCD mixing ratios as the snowpack ages, which increases

the efficacy of BCD SDEs.

Northern subregions are not characterized by the same seasonal correlation between SWE, 2-m temperature, and ISRE. Maximized SWE reductions of 8 mm (4%) occur around 1 June across Greater Idaho, which occurs about a week after the ISRE maximum of (+2.9 W m$^{-2}$) and ~3 weeks before the 2-m temperature maximum (+0.15°C). This peculiarity is due to depressed snowmelt (Figure 2b) through

increased cloudiness (not shown) and precipitation frequency (Figure 2c). Increased snowfall during this period leads to dilution of the snowpack and a depressed ISRE. Additionally, increased precipitation frequencies and attendant cloudiness reduce surface insolation, reducing ISREs further.

The largest overall SWE reductions of 11 mm (5%) occur across the Northern Rockies during mid-June. This period is characterized by a local minimum in 2-m temperature anomalies and local ISRE

values brought forth by increased cloud coverage (not shown). The offsets in absolute SWE reductions with absolute maxima in 2-m temperature and ISRE across the northern subregions are the result of modulations in synoptic-scale weather variability that potentially mask these variables' correlation on time scales finer than weeks.

Figure 10d shows that simulated SDE-induced anomalies in runoff are on the order of millimeters

per day across the four subregions. Maximum simulated precipitation anomalies are generally much less than 0.5 mm d$^{-1}$ (Figure 10c), meaning that runoff anomalies are primarily driven by SDE-induced snowmelt. Here, runoff is defined to be the sum of surface and underground runoff; runoff from glaciers and lakes is neglected. Driven primarily by BC SDEs, runoff is mostly increased through late June across all four subregions. The largest increase in runoff occurs across the Northern Rockies (5.5 mm d$^{-1}$, a 90%

change from CNT), which is characterized by the largest reductions in SWE. During July, negative anomalies in runoff manifest, with the largest reductions simulated across the Northern Rockies (5.5 mm d$^{-1}$, ~5%) and the Utah Mountains (4.5 mm d$^{-1}$, ~75%). Smaller runoff reductions of ~1 mm d$^{-1}$ (2%) are simulated across Greater Idaho, while runoff increases of 1 to 2 mm d$^{-1}$ (< 5%) are simulated across the





Southern Rockies in phase with precipitation increases across this subregion (Figure 10c). Although Qian et

al. (2009) and Wu et al. (2018) emphasized results across basins, the dipole signature of runoff increases

followed by runoff decreases is consistent with our results, even though we primarily examine SDEs at

higher elevations.

SDE-induced perturbations to the weather and hydrological cycle generally stem from

perturbations to the surface energy budget, affecting variables from the "bottom up." BCD ARI on the

other hand seem to affect downwelled irradiance, atmospheric  stability, and clouds in a way that affects the

underlying snow coverage at the surface, effectively altering variables from the "top down." Changes in

clouds and low-level stability were minimal (less than 1% and 0.02°C km$^{-1}$, respectively) and are not

discussed.

**5.2 ARI**

**5.2.1 Spatial patterns of ARI anomalies**

From March through June, BCD ARI-induced 2-m temperature anomalies are minimal (less than

0.1°C in magnitude) across the WUS (Figure 12a).  However, BC and dust seem to impart anomalies of

differing sign across the mountains, with cooling (warming) simulated across the higher elevations due to

BC (dust). This general spatial pattern is correlated with SWE increases due to BC (Figure 12e) and SWE

decreases due to dust (Figure 12f). The cooling and warming patterns associated with BC and dust,

respectively, can be tied to deficits and surpluses in the surface energy budget imparted by these aerosols'

atmospheric radiative effects (to be discussed in section 5.3).

Figure 12d shows that BC ARI-induced SWE increases tend to exceed reductions in SWE

associated with dust ARI. These overall ARI-induced increases in SWE compete against, but do not

exceed, BCD SDE-induced reductions in SWE (Figure 8d). Total simulated precipitation (rain+snow)

anomalies due to BCD ARI are generally less than 0.2 mm d$^{-1}$, while those due solely to snow are less than

0.1 mm d$^{-1}$ (not shown). Additionally, no precipitation variable shows any discernable weekly, monthly, or

seasonal trend. This finding lends further credence to the idea that simulated ARI-induced changes to SWE

manifest from changes to the surface energy budget through BCD ARI.





Figure 12g shows that BCD ARI imparts a negative surface RE. Curiously, BC and dust have surface REs that are opposite in sign (Figures 12h,i). The opposing sign is attributable to the differing microphysical properties of BCD aerosls (discussed in Section 5.3).

**5.2.2 Timing of ARI**

Figure 13a shows the seasonality of competing 2-m temperature anomalies with BC (dust) ARI inducing surface cooling (warming). Maximum BCD ARI-induced temperature perturbations tend to occur from late May into early June. Interestingly, the strongest dust ARI-induced warming (+0.1°C in late May) tends to occur across the Northern Rockies, even though this region lies further away from dust emission sources. As will be discussed, this region tends to have relatively lower burdens of super-micron dust particles compared to the other subregions. As larger (smaller) dust particles tend to absorb (scatter) incoming and upwelling sunlight, this region tends to see more downward-scattered insolation due to the larger fraction of sub-micron dust particles, which results in less surface dimming. BC on the other hand absorbs in the atmosphere, leading to surface dimming and weak surface cooling. Otherwise, the strongest combined (BC+dust) surface cooling is generally simulated to occur across the southern subregions prior to mid-May, but all subregions tend to see BCD ARI-induced cooling throughout the simulation period.

Overall cooling by BCD ARI is accompanied by increased SWE amounts across the four subregions (Figure 13b). Driven by BC ARI, SWE increases of 5 mm (2%) are simulated across the Northern Rockies during mid-May. By percentage however, the largest SWE increases of ~3 mm (~4%) are simulated during the first week of May across the Utah Mountains. Across the southern subregions, the ARI-induced increases in SWE occur earlier in the spring relative to the northern subregions. Meanwhile, the percentage increases in SWE across all four subregions generally increase with time well into the summer, with increases in excess of 10% across the Utah Mountains, the Southern Rockies, and the Northern Rockies by late July (not shown).

Overall, the increase in SWE through ARI is predominately due to BC; however dust ARI does contribute to SWE increases on the order of 1 mm during mid-May and early June across the Utah Mountains and the Southern Rockies, respectively. Interestingly, the Northern Rockies see competition between BC and dust ARI in modulating SWE anomalies. Specifically, BC (dust) increases (decreases)



SWE by 8 mm (4 mm). Differences in these aerosols' optical properties lead to differences in how these

aerosols perturb the surface energy budget while they reside in the atmosphere. These differences are

discussed in Section 5.3; however, it is noteworthy that the changes to snowpack are not due to simulated

ARI-induced changes in precipitation (not shown). Specifically, ARI-induced precipitation anomalies are

similar to those induced by SDE in that they show no discernable trend and generally do not exceed 0.3

mm d$^{-1}$ in magnitude. Because precipitation anomalies are on the order of tenths of millimeters per day, and

SWE anomalies may be more than 10 mm locally, this indicates that ARI-induced changes in SWE are due

mostly to changes in how the snowpack melts with time rather than changes to precipitation.

With SWE increased from April onward due to BCD ARI, runoff tends to decrease across all four

subregions prior to mid-May (Figure 13c). Following 1 June, runoff anomalies become less negative and

even positive across the four subregions, a pattern opposite to that of BCD SDE (Figure 10d). BC ARI tend

to drive a majority of the runoff decreases prior to 1 June and promote increased runoff deeper into the

summer. Dust ARI on the other hand has the opposite effect on runoff to that of BC ARI, increasing runoff

through mid-May and decreasing runoff after 1 June.

**5.3 Main differences between SDE and ARI in affecting snowmelt**

Overall, BC SDE incites anomalies opposite to those of BC ARI. The former heats the snow from

within, introduces a positive radiative perturbation at the surface, and increases melting through surface

albedo reductions. The latter effect preserves the snowpack through aerosol dimming at the surface.

Specifically, BC absorbs direct incoming sunlight, leading to warming of the TOA (Table 3). The

absorption of light by BC within the atmosphere reduces the amount of insolation at the earth's surface,

resulting in a surface RE associated with BC of between -0.5 W m$^{-2}$ and -1 W m$^{-2}$ (see Figure 12h and

Figure 13d; cross symbols). Furthermore, this dimming appears to be maximized across the southern

subregions during mid-May.

Changes to the snowpack brought about by dust ARI and dust SDE are generally smaller than

those brought about by BC SDE and BC ARI. However, a notable instance in which dust ARI brings forth

SWE reductions of 4 mm around 1 June across the Northern Rockies (Figure 13b) can be explained by

considering the microphysical characteristics of dust particles across the subregion. Regional averages of



dust- (and BC-) induced anomalies were computed over areas characterized by high albedos. Smaller dust

particles are generally scattering in nature compared to larger dust particles in shortwave bands (Tegen and

Lacis, 2012). In a single-layer radiative transfer model of the atmosphere, the downward shortwave solar

flux $S^\downarrow$ at the Earth's surface can be expressed as:

$$S^\downarrow = S_0(1 - f_s - f_a),$$    (1)

where $S_0$ is the solar constant at a particular time of day at a latitude characterizing the Northern Rockies

and subject to molecular scattering, $f_s$ is the fraction of $S_0$ scattered by aerosols in the atmosphere, and $f_a$ is

the fraction of $S_0$ absorbed by aerosols in the atmosphere. The upwelled solar flux immediately above the

earth's surface $S^\uparrow$ can thus be found by multiplying (1) by the surface albedo $\alpha$:

$$S^\uparrow = S_0(1 - f_s - f_a)\alpha.$$    (2)

This upwelled radiation will be scattered back and forth between the scattering aerosol layer and the

surface in an iterative process proportional to $f_s$ and $\alpha$:

$$S^\downarrow_{total} = S_0(1 - f_s - f_a)\left(1 + \sum_{n=1}^{\infty} (\alpha f_s)^n\right).$$    (3)

Here, $n$ is the number of times the incident beam is scattered between the surface and the aerosol scattering

layer. $S^\downarrow_{total} - S_0$ is negative. However, by (3), it is clear that in a case where $f_a$ is negligible, $S^\downarrow_{total} - S_0$

becomes less negative as $\alpha$ approaches unity. For atmospheric dust particles residing over the high-albedo

surface of the Northern Rockies, this means that there will be a higher chance of shortwave absorption at

the surface. The reduced negativity of the dust surface shortwave RE, *together* with the dust longwave

warming (Figure 12i and Figure 13d), explains why dust aerosols contribute to snowpack reductions across

the Northern Rockies. The physical process described here is similar to that noted in Stone et al. (2008)

who examined the atmospheric REs of wildfire smoke across northern Alaska's high albedo surface.


### 6. Consequences of increased dust

The mean [dust] was undersimulated in all WRF-Chem experiments by 63% compared to

IMPROVE (Section 3.2). Even in the DTF=2 experiment, [dust] was still underpredicted by 43%. Figure

14 shows the consequences of increasing dust emissions by 60% from March through June. ISRE is

increased by 0.2 W m$^{-2}$ in many areas, especially across the southern subregions. Additionally, the northern



subregions see ISRE increases of 0.05 to 0.1 W m$^{-2}$ (Figure 14a). Across Greater Idaho and the Northern Rockies, this represents a doubling of the ISRE in the DTF=2 experiment compared to CNT.

As discussed in the previous section, in-atmosphere dust can warm the surface through shortwave and longwave ARIs over high-albedo surfaces. Figure 14b shows the difference in the net surface RE

between the DTF=2 experiment and CNT. A positive differnce of between 0.01 and 0.05 W m$^{-2}$ is prevalent across the four subregions, especially across the Northern Rockies and Southern Rockies. Immediately downstream of major dust emissions sources, such as the Mojave Desert and those of central Montana, we see enhanced surface dimming in excess of 0.2 W m$^{-2}$ in DTF=2, as larger (more absorbing) dust aerosols reduce insolation at the surface.

Enhanced warming of the surface, both through dust ARI and dust SDEs, leads to widespread SWE reductions in DTF=2 compared to CNT (Figure 14c). These reductions exceed 5 mm in many areas and are generally accompanied by small temperature increases (Figure 14d) of less than 0.05°C. The seasonality of SWE differences between DTF=2 and CNT are shown in Figure 14e. It is clear that increasing dust emissions has consequences for snowmelt across our four subregions, especially across the

Northern Rockies. Through enlargements in the dust SDE and dust ARI, SWE reduction enhancements of 7.5 mm are simulated in DTF=2 compared to CNT across the Northern Rockies. These melting enhancements by themselves are ~70% as large as the reductions in snowpack simulated due to the BC SDE. BC SDE-induced SWE reductions across the Northern Rockies are the largest of any subregion and larger than any other effect across any of the four subregions. Therefore, it is possible that the dust SDE

and dust ARI are actually comparable to those induced by BC SDE.

**7. Combined effects of BCD**

Figure 15 shows time series of the total radiative (SDEs+ARI) impacts of BCD on the four subregions. These aerosols combine to heat the surface temperature by up to 0.15°C, with the largest

heating across the Utah Mountains in mid-spring and across the Northern Rockies during early summer (Figure 15a). The largest BCD SDE-induced SWE reductions occur across the Southern Rockies (10 mm) and Northern Rockies (12 mm) during mid-May and mid-June, respectively (Figure 15b). As mentioned


earlier, changes in SWE are predominately driven by SDEs, although dust ARI can effectuate SWE reductions, especially across the Northern Rockies.

With the exception of the Utah Mountains, BCD-induced perturbations in SWE modulate runoff increases in the late spring and early summer, decreasing runoff later in the summer (Figure 15c). Maximum runoff changes are simulated across the Northern Rockies, with increases (decreases) of 5 mm d$^{-1}$ (5 mm d$^{-1}$) occurring in mid-May (late July). Despite the Utah Mountains being located closest to BC and dust emission sources, its SWE and runoff changes are not as large as those across the Southern Rockies

and Northern Rockies. This may be due to the close positioning of the Utah Mountains relative to the southwestern deserts. Larger, more dust aerosols dim sunlight, effectuating a negative surface RE. Combined with BC ARI dimming, the dust ARI offsets snow darkening reductions in SWE, resulting in relatively smaller net perturbations to the surface water budget.

**8. Conclusions**

Using seven "branch" WRF-Chem experiments with a horizontal resolution of 4 km, the SDEs of BCD were quantified across four subregions of the WUS for water year 2009. These aerosols' ARI were also examined and quantified.

It was found that BC surface concentrations in the WRF-Chem control experiment (CNT) were

well simulated compared to observations from IMPROVE, while dust surface concentrations [dust] were undersimulated by 63%. An additional simulation was run with domain-resolved dust emissions increased by 60% (DTF=2), but this simulation still underpredicted [dust] by a factor of 43% compared to IMPROVE. It was found that CNT generally overpredicted precipitation compared to PRISM, snow cover compared to MODIS, and SWE compared to UA at all elevations; the wet bias increased with increasing

elevation. Meanwhile, CNT simulated a warm (cold) bias at lower (higher) elevations. The biases/elevation Pearson values exceeded +0.3 for hydrologic reference variables and fell below -0.65 for the temperature variables.

It was found that BCD SDEs generally reduce springtime and summertime SWE while ARIs generally increase SWE during this same time period. However, SWE reductions due to the SDE, typically

on the order of 2% to 5%, far exceeded the increases induced by ARIs. SWE changes by these BCD effects



led to changes in runoff. Specifically, SDEs bring forth runoff increases of 1-5 mm d$^{-1}$ before July and runoff reductions of a similar magnitude during July. Runoff increases and decreases were largest across the Northern Rockies, with runoff increases of 95% during springtime and early summer preceding gradual decreases of 5% through the first half of July. BC ARI on the other hand dims incoming solar radiation by

0.5-1.0 W m$^{-2}$, depressing snowmelt in the late spring. The resulting ARI-induced perturbation to runoff is generally opposite to that associated with SDE. To emphasize, the changes to surface hydrology across the WUS are driven primarily by SDE, not ARI, and simulated SWE anomalies brought forth by BCD were not due to precipitation anomalies.

It was found that BC ARI dims the surface, resulting in a negative surface RE. Depending on the

subregion however, dust ARI could actually incite a positive surface RE. We believe this to be the result of differing optical properties associated with dust aerosols of differing size. Across subregions in relatively close proximity to dust emissions sources, larger, more absorbing dust particles dim the surface, preserving snowpack through a negative surface RE. Across regions further away from dust emission zones, namely the Northern Rockies, dust particles tend to be smaller and more scattering. Across this high-albedo

surface, downwelled solar radiation is reflected upward and subsequently backscattered to the surface by the scattering dust layer in an iterative process. Suppressed dimming of incoming solar energy, in tandem with a positive longwave RE due to dust ARI, warmed the surface and reduce SWE.

Indeed, BC-induced effects on WUS meteorology and hydrology for water year 2009 were generally simulated to be larger than those induced by dust. We conclude however by noting that there

were non-negligible changes in SWE in the instance that dust emissions were increased. Specifically, SWE reduction enhancements of several millimeters were simulated. While our results were consistent with Wu et al. (2018), we note that observed [dust] was estimated following Kevouras et al., (2007) and this estimation is subject to uncertainties therein. There remains a possibility that dust effects may be quite comparable to those of BC, although further simulations are required to answer this question. More

generally, BCD SDEs and ARI can impart significant perturbations of WUS weather and hydrology and corroborate the results of coarser resolution GCMs. Future studies should focus on increasing the domain size in order to quantify larger synoptic-scale circulation changes associated with BCD effects; this coupling has been noted in previous GCM experiments (e.g. Rahimi et al. 2019), but the domain size used



in this study was too small to resolve these potentially significant larger-scale responses. Additionally,

because convection was not parameterized, an opportunity exists to quantify how orographically forced

precipitation events are impacted by BCD SDE. Specifically, this output data can be used to quantify how

updraft speeds, convective updraft area, and storm energetics change as a function of BCD effects, which

will be topics of future study.

**Author contributions**

S. Rahimi conducted the WRF-Chem experiments and performed analyses. X. Liu provided guidance in

the scientific conceptualization. Z. Lu, Z. Lebo, and C. Zhao assisted in certain technical aspects of WRF-

Chem.

**Acknowledgements**

We acknowledge the overseers of IMPROVE, a collaborative association of state, tribal, and federal

agencies, and international partners. US Environmental Protection Agency is the primary funding source,

with contracting and research support from the National Park Service. The Air Quality Group at the

University of California, Davis is the central analytical laboratory, with ion analysis provided by Research

Triangle Institute, and carbon analysis provided by Desert Research Institute. We also thank Stu McKeen

for his help with the use of his emissions generation software.

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

**Tables**

**Table 1. Listing of WRF-Chem specifications. NOCHEM is identical to CNT, but without the chemistry options.**

|  | Scheme/Option | Reference |
|---|---|---|
| **Chemistry** | | |
| Aerosol model | MOSAIC 4-bin with aqueous reactions | Zaveri et al. (2008) |
| Photochemical model | CBM-Z | Zaveri and Peters (1999) |
| Dust emissions | GOCART | Ginoux et al. (2001) |
| Biogenic emissions | Guenther | Guenther et al. (1993) |
| Fire emission | FINN 1.5 | Wiedinmyer et al. (2011) |
| Anthropogenic emissions | EPA-NEI11 | U.S. EPA |
| **Physics** | | |
| Forcing data | CFSR | Saha et al. (2010) |
| Microphysics | Morrison double-moment | Morrison et al. (2009) |
| Radiation | RRTMG | Iacono et al. (2008) |
| Land surface model | CLM4 | Oleson et al. (2010) |
| Boundary layer | Yonsei University | Hong et al. (2006) |
| Surface layer | Monin-Obukhov | Zhang and Anthes (1982) |

**Table 2. Listing of WRF-Chem experiments organized by the types of BCD effects included in each simulation.**

| Experiment | BC SDE | BC ARI | Dust SDE | Dust ARI |
|---|---|---|---|---|
| CNT | yes | yes | yes | yes |
| noSDE | no | yes | no | yes |
| noARI | yes | no | yes | no |
| noBCD | no | no | no | no |
| noBCSDE | no | yes | yes | yes |
| noBCARI | yes | no | yes | yes |






**Table 3. Monthly averages of BCD radiative effect (W m$^{-2}$) at the TOA across the domain and subregions therein. Dust RE values are given in parentheses.**

| BC (dust) TOA clear-sky radiative effect (W m$^{-2}$) | | | | | |
|---|---|---|---|---|---|
| Month | Domain | Greater Idaho | Northern Rockies | Utah Mountains | Southern Rockies |
| February | 0.32 (-0.67) | 0.30 (-0.50) | 0.31 (-0.58) | 0.44 (-0.70) | 0.36 (-0.62) |
| March | 0.59 (-0.99) | 0.62 (-0.69) | 0.65 (-0.83) | 0.75 (-0.92) | 0.68 (-0.81) |
| April | 0.67 (-1.04) | 0.74 (-0.92) | 0.79 (-0.96) | 0.81 (-1.02) | 0.80 (-0.89) |
| May | 0.60 (-1.08) | 0.59 (-0.72) | 0.65 (-0.92) | 0.62 (-1.11) | 0.65 (-1.05) |
| June | 0.58 (-1.08) | 0.47 (-0.92) | 0.57 (-0.90) | 0.51 (-1.01) | 0.50 (-0.67) |
| July | 0.52 (-0.99) | 0.38 (-0.63) | 0.40 (-0.90) | 0.43 (-0.86) | 0.36 (-0.80) |










**Table 4. Same as in Table 3, but for in-snow BCD radiative effect**

| BC (dust) surface in-snow radiative effect (W m$^{-2}$) | | | | |
|---|---|---|---|---|
| Month | Greater Idaho | Northern Rockies | Utah Mountains | Southern Rockies |
| February | 0.15 (0.05) | 0.24 (0.04) | 0.25 (0.12) | 0.29 (0.09) |
| March | 0.21 (0.06) | 0.34 (0.12) | 0.54 (0.43) | 0.54 (0.23) |
| April | 0.47 (0.12) | 0.58 (0.13) | 0.73 (0.42) | 0.75 (0.25) |
| May | 0.73 (0.18) | 1.09 (0.29) | 0.52 (0.28) | 0.75 (0.29) |
| June | 0.56 (0.13) | 0.93 (0.22) | 0.20 (0.08) | 0.41 (0.13) |
| July | 0.17 (0.04) | 0.57 (0.13) | 0.08 (0.03) | 0.16 (0.04) |









**Figures**

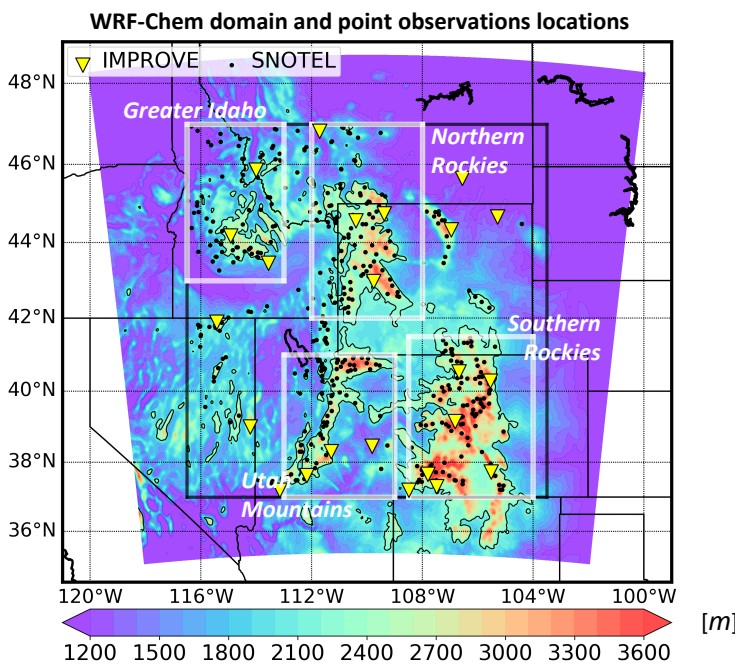


**Figure 1. WRF-Chem domain with analysis subregions (white transparent boxes). The colorfill represents the surface elevation (m), and the thin black line denotes the low-pass filtered 2,200-m isopleth. The thick black line bounds our analysis region. Black circles represent the 424 analysis SNOTEL sites, while the yellow inverted**
**triangles represent the 29 analysis IMPROVE sites.**

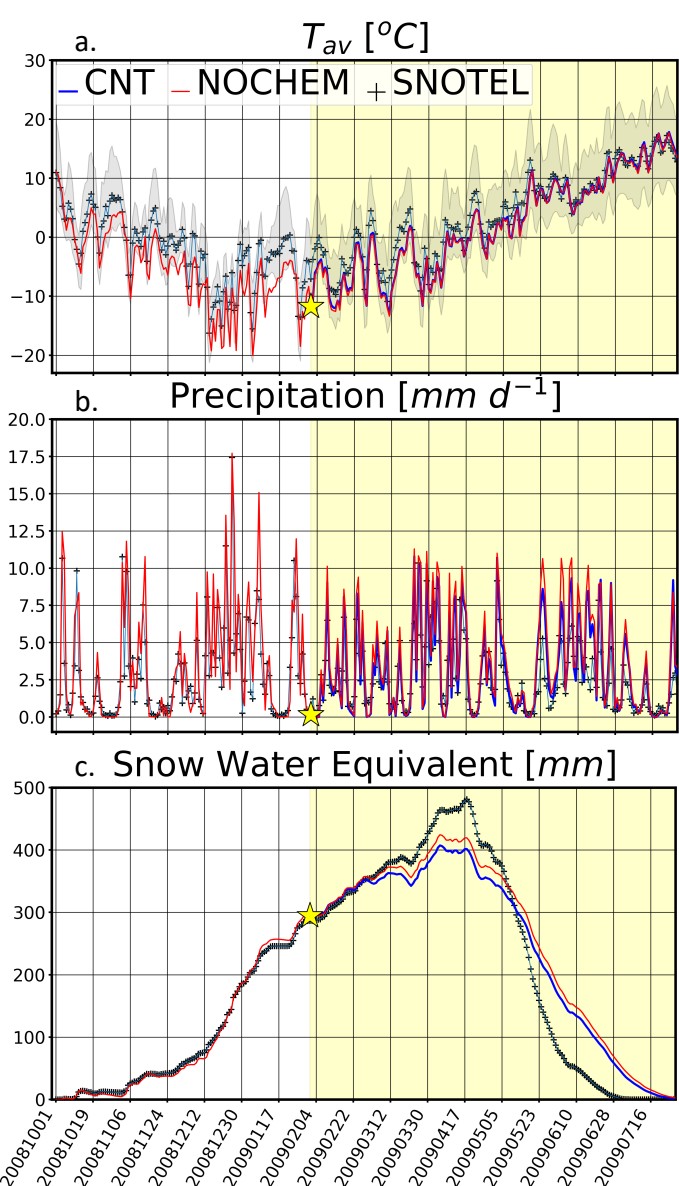

**Figure 2. Time series of CNT, NOCHEM, and SNOTEL daily (a) $T_{av}$, (b)
precipitation rate, and (c) SWE. The gray colorfill in (a) spans the range of
SNOTEL-observed $T_{min}$ and $T_{max}$. The yellow star denotes the launch point of all
CHEM simulations (1 February 2009 at 00:00 UTC), and the soft yellow shading
highlights our period of interest (1 February 2009 through 1 August 2009).**

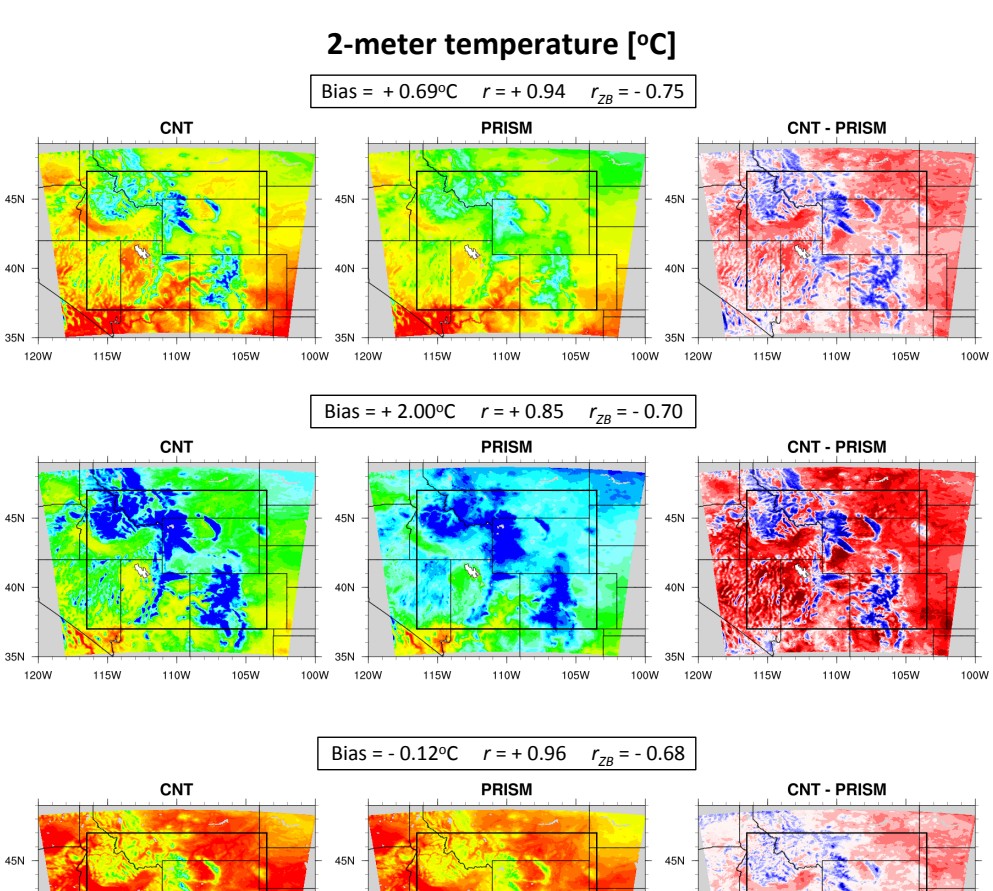

**Figure 3.** $T_{av}$ (top row), $T_{min}$ (middle row), and $T_{max}$ (bottom row) averaged from February through July. CNT results are shown in column 1, PRISM results in column 2, and CNT – PRISM (bias) results are shown in column 3. The black box denotes the region of the domain in which biases and other statistics are computed. Mean bias, $r$-value, and $r_{ZB}$-value are given for each variable.





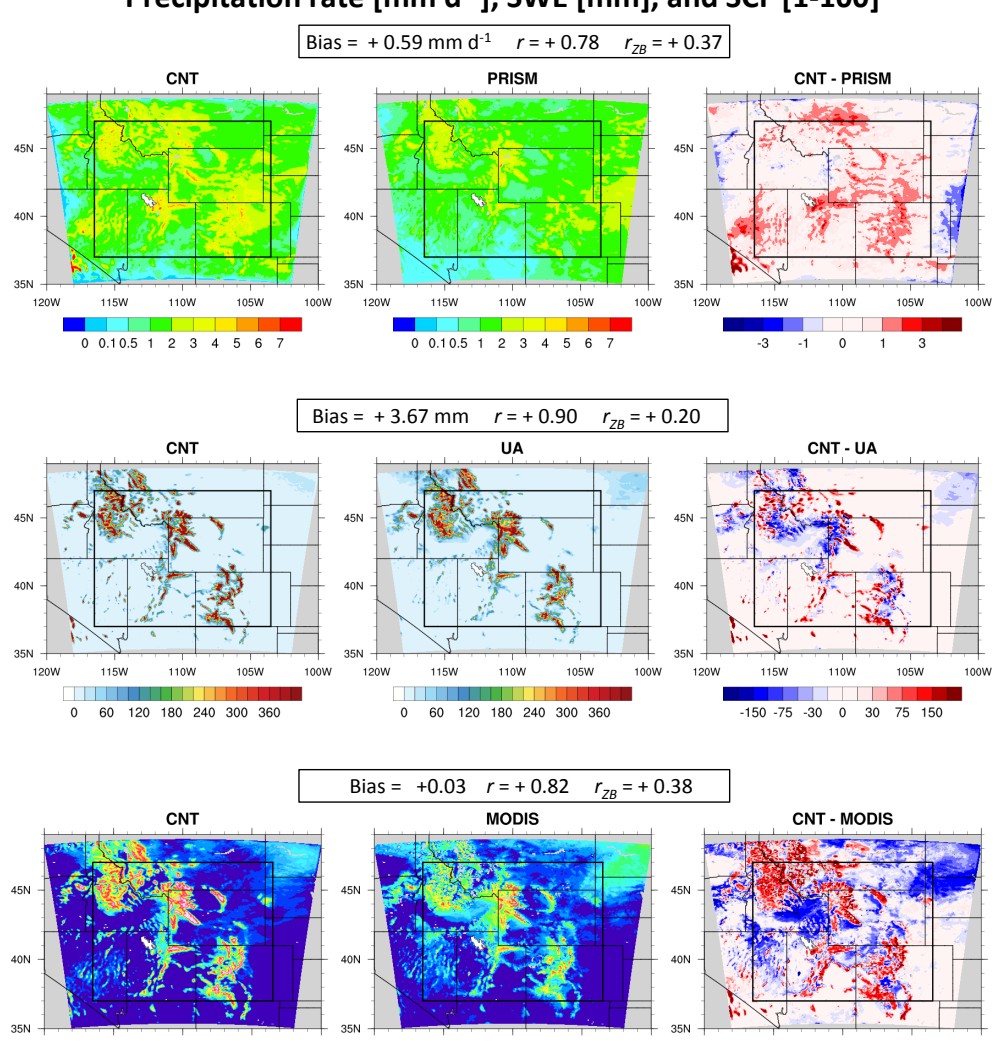

**Figure 4. As in Figure 3, but precipitation rate is compared to PRISM (top row), SWE is compared to UA (middle row), and SCF is compared to MODIS (bottom row).**

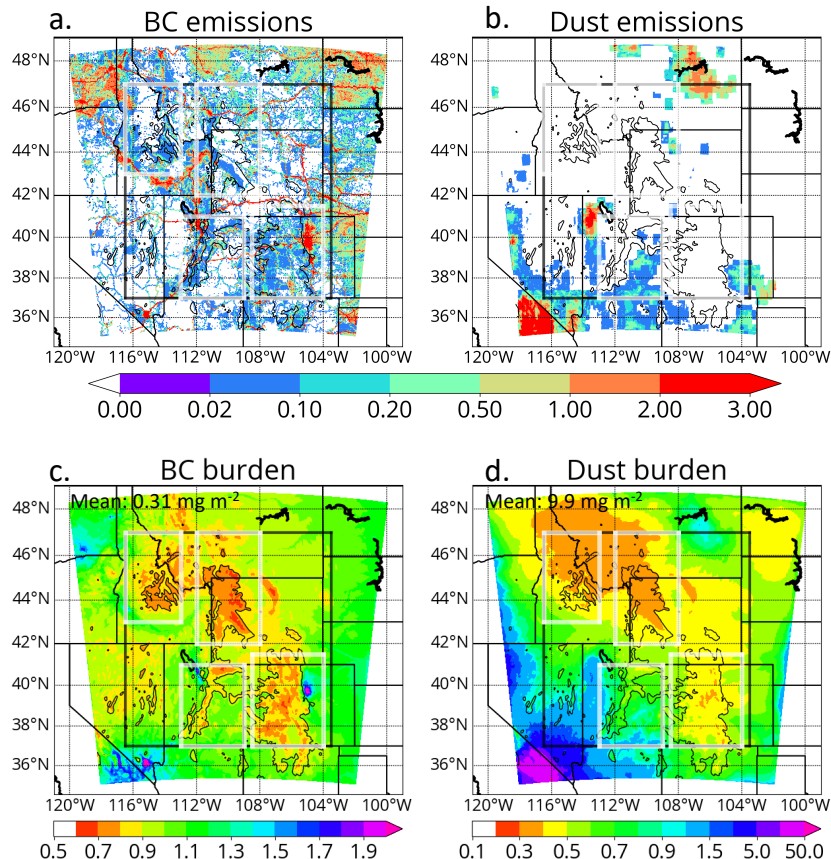

**Figure 5.** **Spatial distribution of (a) February-through-July-averaged anthropogenic BC emission rates from EPA NEI ($\mu g\ m^{-2} s^{-1}$) and (b) integrated dust emission rates ($\mu g\ m^{-2} s^{-1}$). Panels (c) and (d) show the February-through-July-averaged mean-normalized BC and dust burdens, respectively (unitless). Note the difference in the colorbars.**

985





**Figure 6. Time series of BC (a) and dust (b) from CNT and IMPROVE. Red transparent circles denote individual IMPROVE measurements. The lower panel (c) shows the same data as in the top and middle panels, except the data are temporally averaged from February through July instead of spatially averaged.**

990



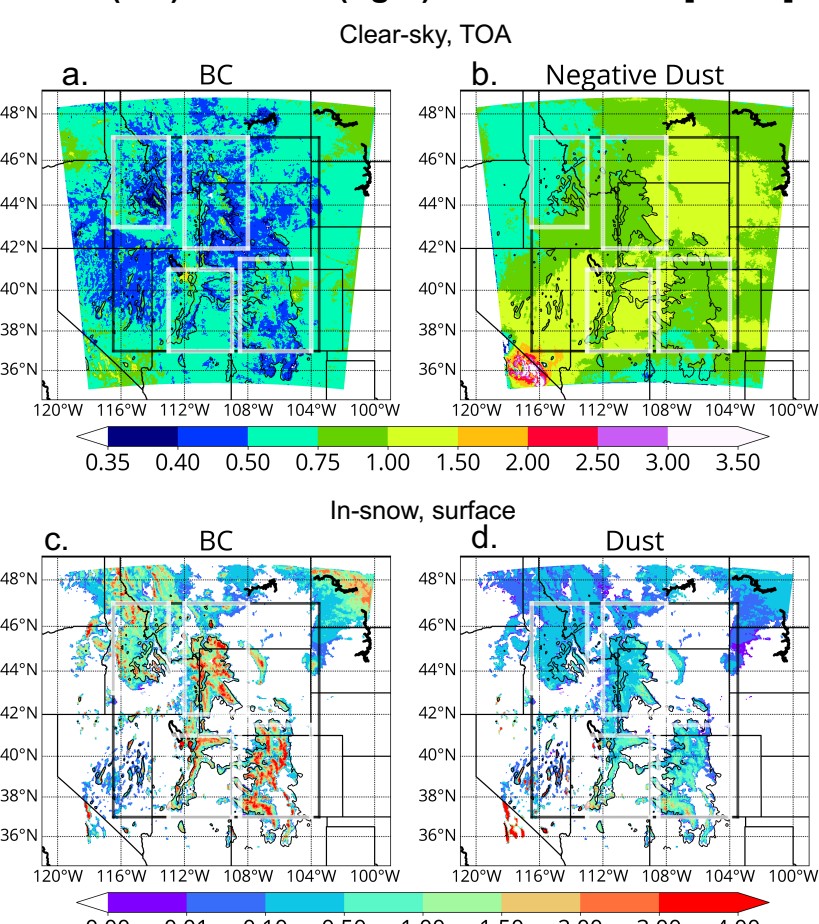

**Figure 7. Top-of-the-atmosphere (TOA) clear-sky (top) and in-snow (bottom) radiative effect for (left) BC and (right) dust averaged from February through July (W m$^{-2}$). The TOA radiative effect panels (a and b) make use of the 00, 06, 12, and 18 UTC output files, while the in-snow radiative effect panels (c and d) make use of the 18 UTC output files only.**



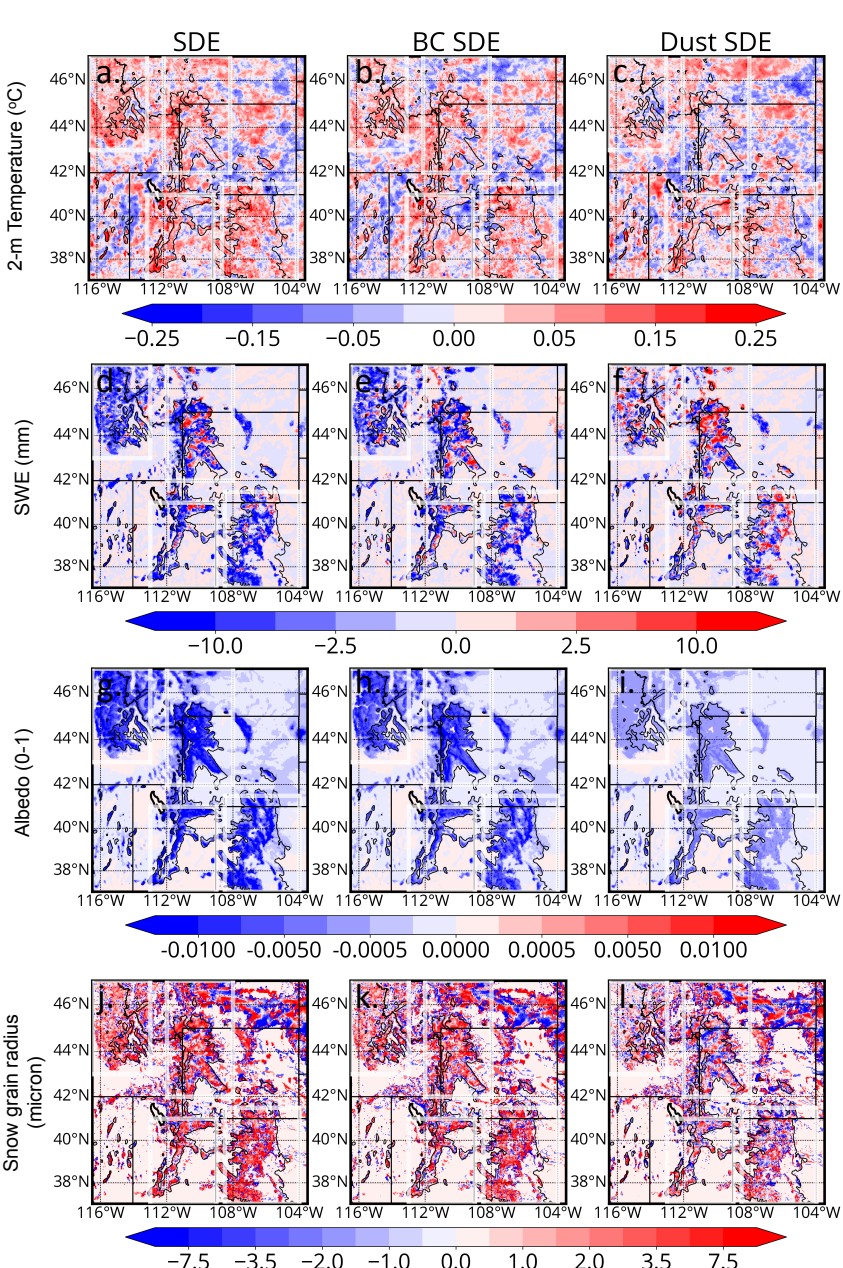

**Figure 8.** March through June averaged SDE-induced anomalies in (top row) 2-m temperature, (second row) SWE, (third row) albedo, and (bottom) snow grain effective radius in the top snow level for (first column) BC+dust, (second column) BC, and (third column) dust. The thin black contours denote the 2,200 m elevation contour.





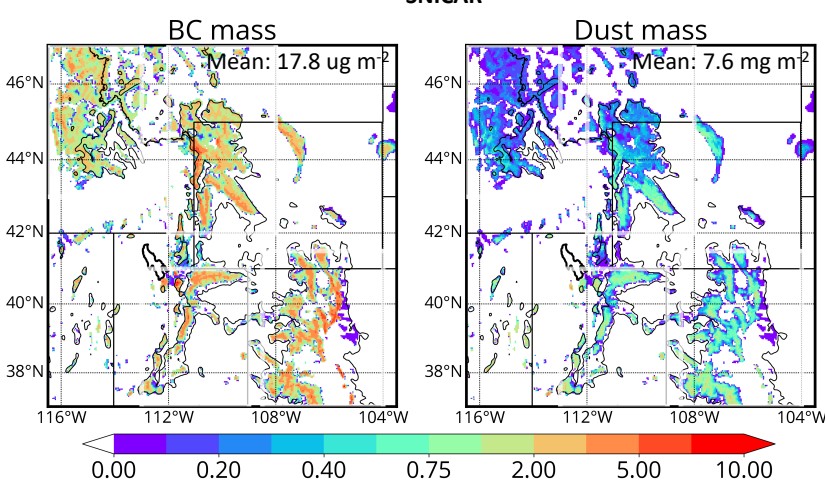

**Figure 9. Top layer in-snow burdens for (left) BC and (right) dust normalized by their respective means (unitless). Means are computed for gridcells that have SWE values greater than 2 mm in the March through June average.**






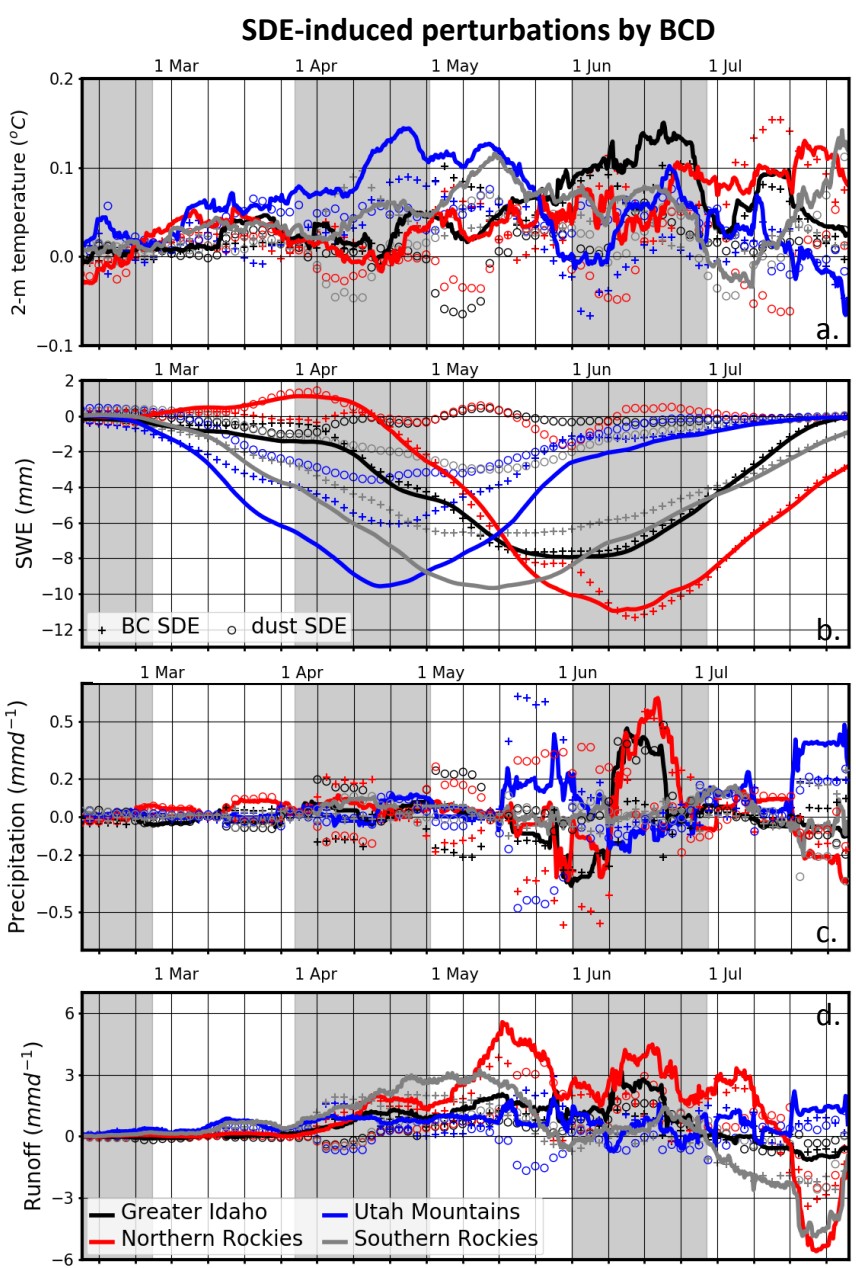

**Figure 10. Presented by region are low-pass filtered time series of perturbations in (a) 2-m temperature, (b) SWE, (c) precipitation, and (d) runoff incited by BCD SDE. Solid lines show perturbations due to total (BC+dust) SDEs, while crosses (hollow circles) show perturbations due to BC (dust) only.**




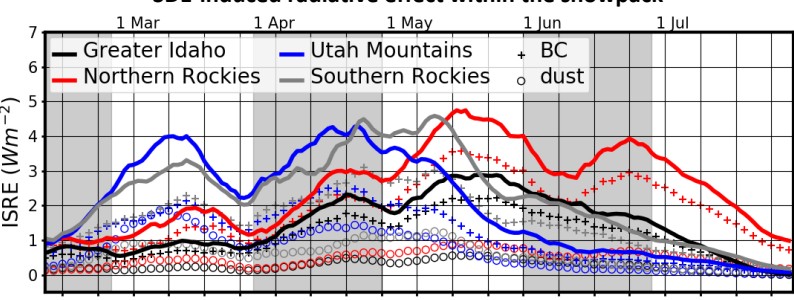


**Figure 11. Presented by region are low-pass filtered time series of surface ISRE across various subregions, separated by aerosol type (BC or dust).**











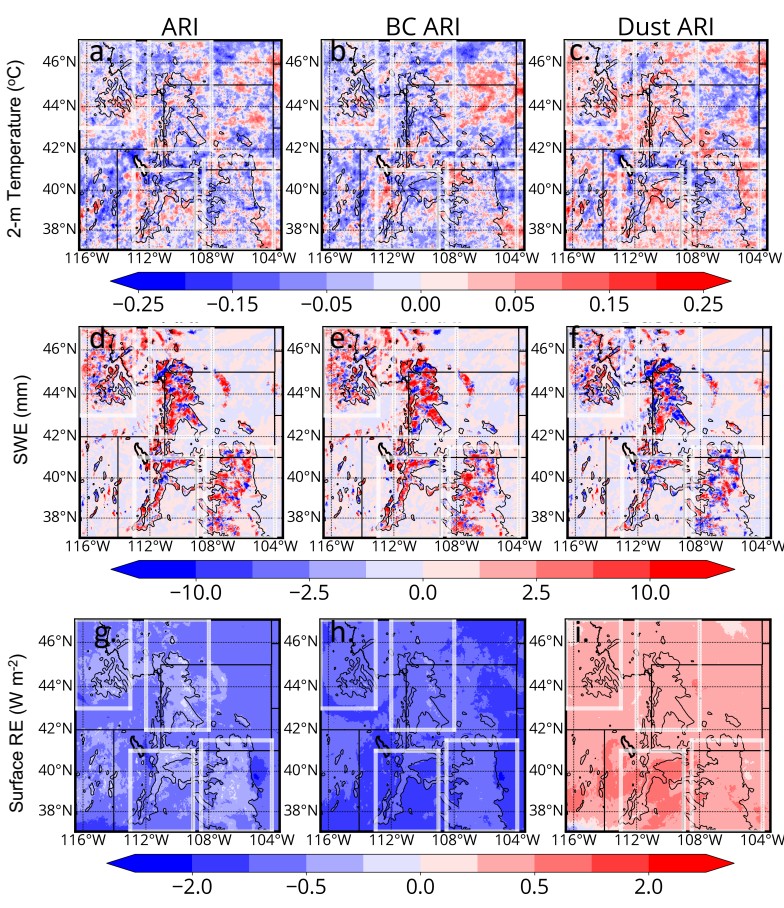

**Figure 12. Same as in Figure 8, but for (top row) 2-m temperature, (middle row) SWE, and (bottom row) surface radiative effect (RE) due to (g) BC+dust, (h) BC only, and (i) dust only. RE values are computed diagnostically following Ghan et al. (2012).**

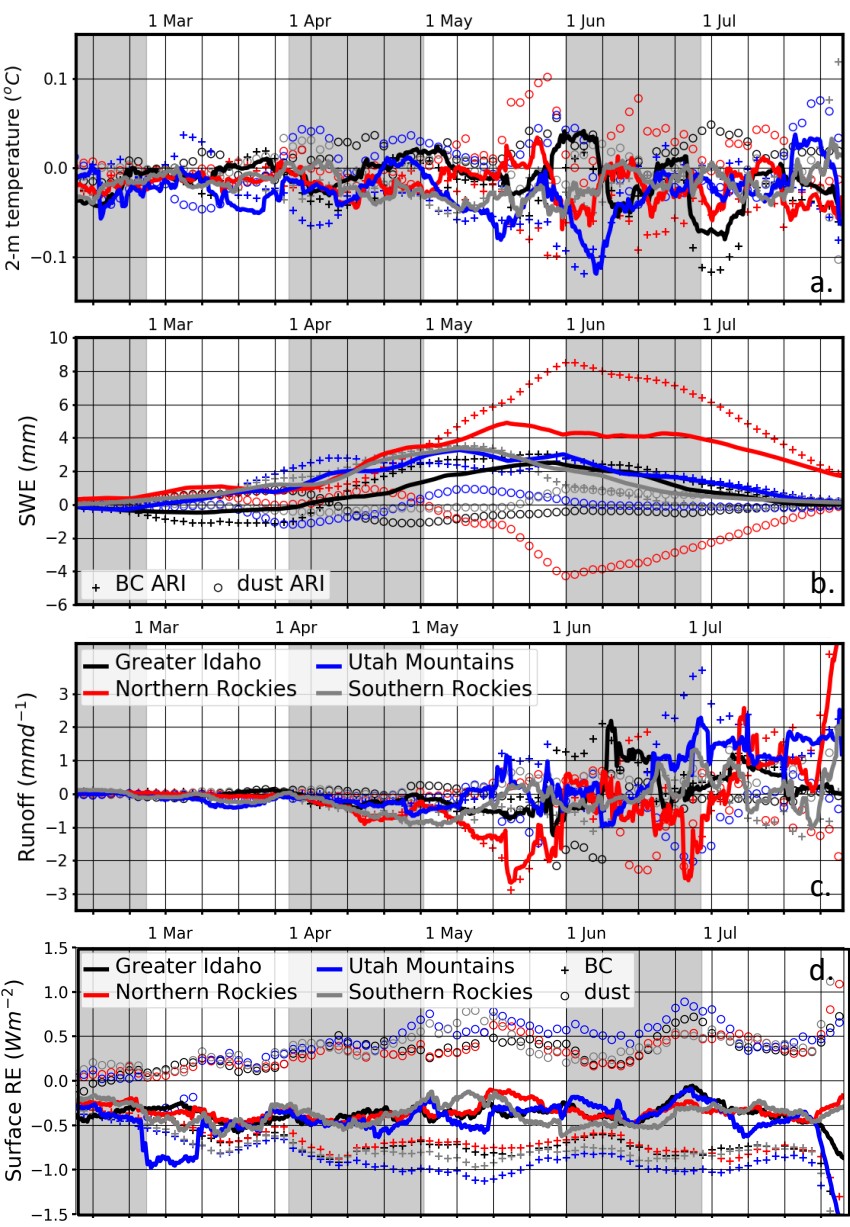

**Figure 13. Same as in Figure 10, but for ARI-induced perturbations. Additionally, panels (c) and (d) show ARI-induced perturbations in runoff and the surface energy budget. Crosses in panel (c) show BC-induced perturbations to the surface energy budget, while hollow circles show dust-induced perturbations.**




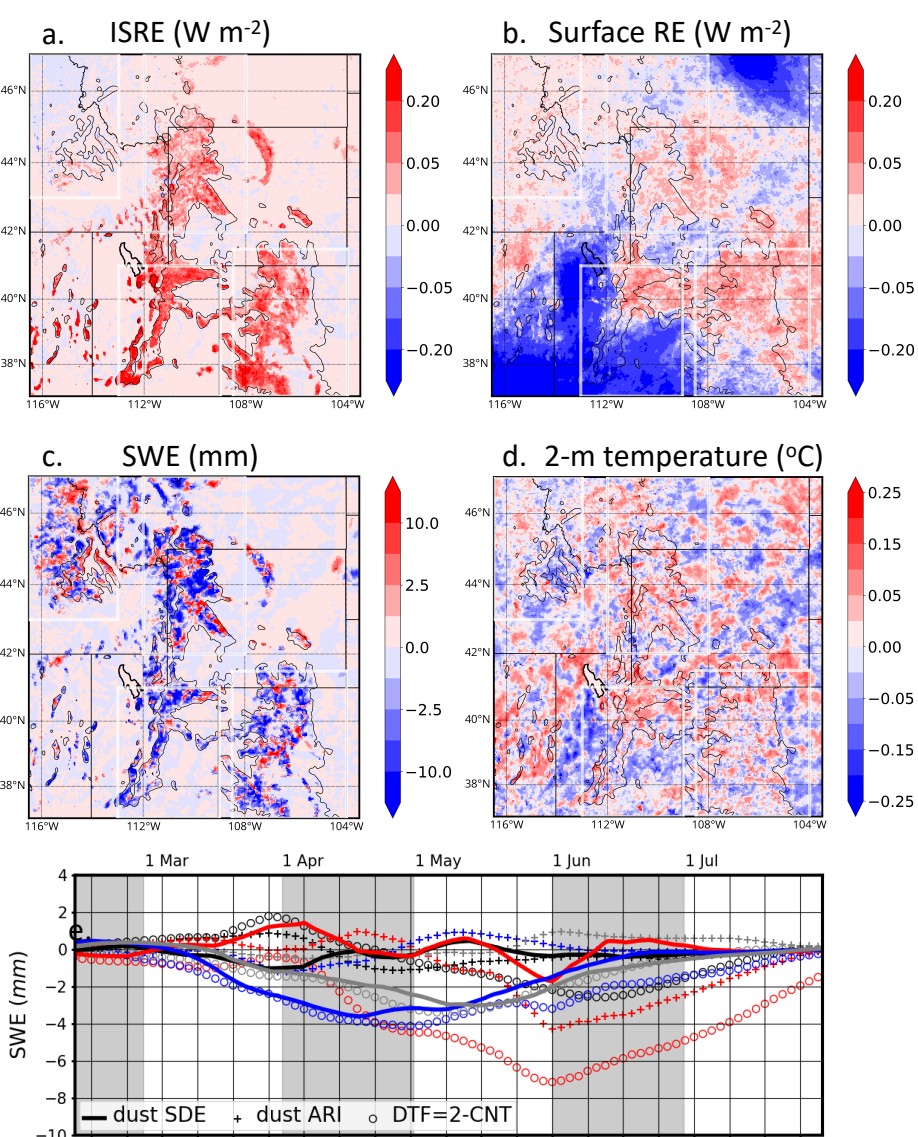

**Figure 14.** DTF=2 – CNT March-through-June averaged (a) ISRE, (b) surface RE, (c) SWE, and (d) 2-m temperature. Panel (e) shows low-pass filtered time series of dust SDE from CNT (solid lines), dust ARI from CNT (crosses), and perturbations to SWE brought forth by increasing the DTF to 2 (DTF=2 – CNT; hollow circles). Colors indicate the subregions, as in previous figures – Black (Greater Idaho), Red (Northern Rockies), (Blue) Utah Mountains, and (Gray) Southern Rockies.


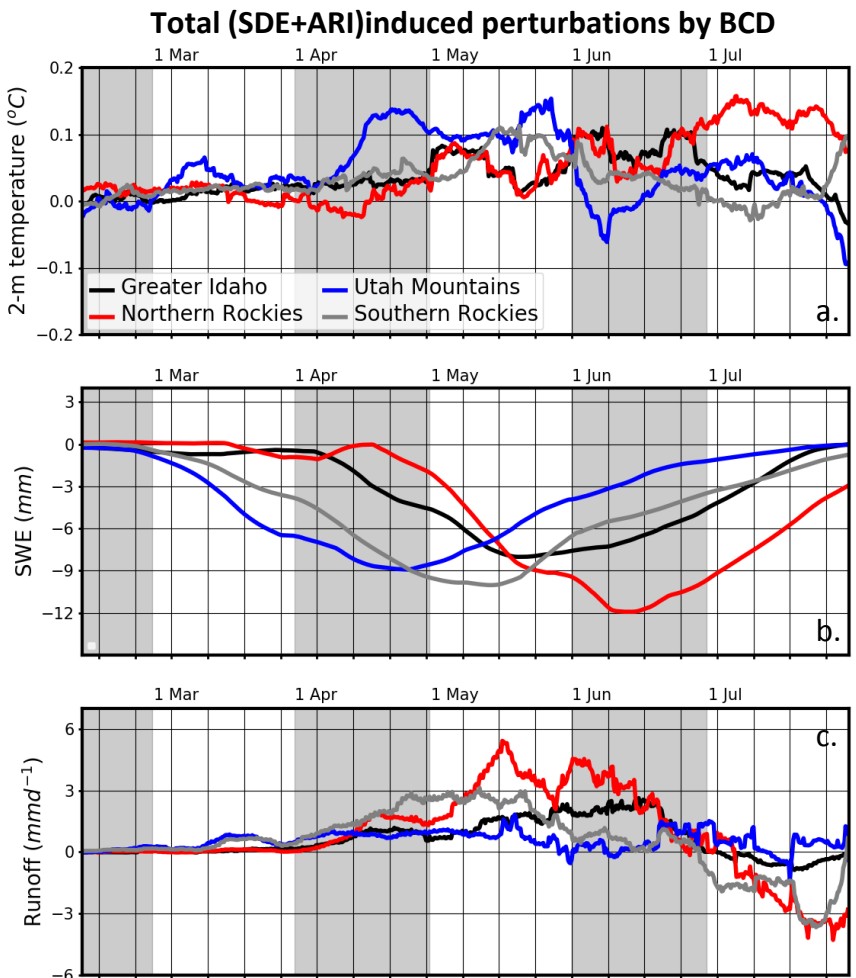

**Figure 15. Total (SDE+ARI) region-specific perturbations to (a) 2-m temperature, (b) SWE, and (c) runoff induced by BCD.**