# Peer review of "Examining the atmospheric radiative and snow-darkening effects of black carbon and dust across the Rocky Mountains of the United States using WRF-Chem"

_Atmospheric Chemistry and Physics, 2019_

## Referee Comment (RC1) · Anonymous Referee #1 · 26 Jan 2020

The authors present a modeling study on the impact of black carbon (BC) and dust on the regional climate of the Rocky Mountains. Using WRF-Chem they examined the radiative impact of BC and dust in the atmosphere and the impact on the snow pack via the modification of snow albedo and snow melting. They performed a series of simulations with all processes or with some processes eliminated. WRF-Chem simulations were limited to the period from February to July 2009 after spin-up simulations with WRF without chemistry. The simulations give important information on the contrasting radiative impacts of BC and dust in the atmosphere and the snow. For example, they

confirm the larger radiative impact of BC compared to dust despite the orders of magnitude higher concentration of dust. The results also demonstrate the different regimes in four specific regions of the Rocky Mountains. The authors continue to discuss the potential impact of BC and dust on hydrological processes and specifically on the timing of the run-off. I have major concerns concerning this part of the manuscript, which in my opinion is less developed and less convincing. Therefore, I recommend major revisions before publication of the manuscript in ACP.

Major comments: In the simulations the CNT run only kicks in after 01/02/09. However, at this date approximately 60 % of the snow has on average already been deposited (Fig. 2c), but the BC and dust loading of this part of the snowpack is not known from the NOCHEM runs. How is this treated? Does the snowpack consist of a lower part of clean snow with layers including BC and dust on top? If yes, what is the impact on the simulations? How was this taken into account for calculated parameters (e.g. for the BC and dust in-snow burdens)?

The authors claim in ch. 5.2 that since changes in simulated precipitation are at most 0.3 mm d-1 and SWE anomalies can be larger than 10 mm, the induced changes in SWE cannot be attributed to precipitation changes. I don't find this a convincing argument. Assuming that early in the winter season the solid precipitation increased by the given upper limit only for a period of a month and if all further processes remain unchanged, the resulting SWE for the rest of the winter season would increase by 9 mm. Therefore, the impact of precipitation changes in the simulations should be analyzed and discussed in more detail.

In general, the seasonal SWE average is in my opinion not an appropriate parameter since it includes the history of the precipitation. Solid precipitation early in the winter season has a larger impact on the average than later precipitation. The same is true for the simulation: if the SWE is modified early in the simulations the impact on the SWE average is larger than for later modifications. This leads than to confusing statements that the simulated SWE is larger than the observed SWE (e.g. ch 3.2), while Fig. 2c

clearly show lower maximum SWE values in the NOCHEM and CNT runs. The positive bias probably stems only from the delayed snow melting in the simulations. Maybe, anomalies are better analyzed using SWE values at several specific dates? This could show the negative bias in spring and the positive bias in summer in the simulated SWE.

Fig. 2c demonstrates further that the dynamics of the snow melting are not reproduced by the model independent if it includes BC and dust or not. Including BC and dust seems to shift the melt-out dates of the snow by a couple of days, but the simulated melt-out still appears to be delayed on average by more than 20 days compared to the observations. Moreover, observed melting rates are significantly higher than simulated melting rates. This should be discussed in more detail. This bias leads for example to large simulated impacts of BC and dust in the snow on temperature, SWE and run-off in July, for which the observations show no or rather little snow on the ground.

In the manuscript the hydrological impact is directly linked to surface run-off related to snow melting, without taking into account any detailed hydrological processes like groundwater storage or sub-surface transfer. This should be mentioned in the manuscript and potential impacts should be discussed. Moreover, since the dynamics and the timing of the snowpack melting in the simulations do appear to be biased (see above), it appears likely that the derived run-off is also strongly biased. How reliable are the conclusions concerning shifts in the timing of the run-off? A comparison with observed run-off data like for the atmospheric and snow data would be very helpful to support the conclusions in this part of the manuscript. In my opinion, related to this bias the simulations can at most give relative changes according to run off shifts in the runs with and without BC and dust. In my opinion, the presented shifts in run-off are not realistic and can in its current form not be used to inform local stakeholders. I recommend deleting from the manuscript all results and further parts describing and discussing the derived run-off.

Minor comments: Concerning the impact of a modified snow pack on the hydrology of the western part of the US the authors refer in the introduction to Serreze et al., 1999

and Hamlet et al, 2007, which are both based on data from the last century. Adding studies on this subject based on more recent observations would be valuable for the readers.

It appears that the used emissions covered 2011, while the simulations covered the first half of 2009. It remains unclear for which year the boundary conditions are valid. Any specific conditions during any of the considered years? The potential impact should be briefly discussed.

It would be good to recall in ch. 2.1 how the introduction of BC and dust into the snowpack due to dry and wet deposition is treated in the SNICAR model and if and how BC and dust are preserved in the snow during melting.

In Fig. 2c it appears that the only significant difference between averaged SWE in CNT and NOCHEM occurs in the first half of March. Afterwards, the two curves seem to behave very similar with more or less constant differences. Is the impact of BC and dust in the snow on the simulated SWE only apparent in this short period? For example, the authors could show in Fig. 2c also the difference in SWE from CNT and NOCHEM to clarify this. I would actually expect that the impact is stronger during the melting phase than in March. If this is not the case, this should be discussed.

The data shown in Fig. 2c cover a huge area. It would be useful to show the same curves also for the four selected regions, which exhibit in the simulations different snow dynamics as discussed later on the manuscript. Are there similar differences in observed and simulated SWE in the four specific regions?

The description of the impact of BC on snow metamorphism in lines 383ff appears rather superficial. The presence of absorbers in the snow has multiple impacts on the properties of the snow, which finally contribute to the radiative forcing. More detailed descriptions of the processes can for example be found in Painter et al., 2007 and Flanner et al., 2007.

References Flanner, M. G., Zender, C. S., Randerson, J. T., and Rasch, P. J.: Present-day climate forcing and response from black carbon in snow, J. Geophys. Res., 112, D11202, doi: 10.1029/2006JD008003, 2007. Painter, T. H., Barrett, A. P., Landry, C. C., Neff, J. C., Cassidy, M.P., Lawrence, C. R., McBride, K. E., and Farmer, G. L.: Impact of disturbed desert soils on duration of mountain snow cover, Geophys. Res. Lett., 34, L12502, doi: 10.1029/2007GL030284, 2007.

---

## Referee Comment (RC2) · Anonymous Referee #2 · 30 Apr 2020

Based on model simulations, the authors examine the skill of high resolution WRF-Chem on the impact of snow albedo darkening and radiative forcing over western USA. They evaluate the model simulation with various observations. The authors also discussed the radiative differences between BC and dust and intercompared two different pathways (snow direct radiative effect and atmosphere direct radiative effect) of aerosol effect on surface water budget. The spatial and temporal variation of radiative effects are also discussed. The experiments and results are interesting and suitable for publication in ACP after major revision to address following concerns.

[Figure]

1) More discussion on role of dust in the manuscript is needed along with clear explanation and analysis. A)why is dust induced SWE positive over Northern Rockies in Figure 8f,which is inconsistent with the fact that dust-ISRE is positive (Figure 7d) and dust-induced change in albedo is negative (Figure 8i). B) More aerosols always have a negative radiative effect at surface as it either scatters or absorbs incoming radiation at surface. Why is dust surface RE in Figure 12i positive? Although, the authors have tried to explain this by stating the differences in aerosol microphysics, I feel it is not clear. C) Also, this is found over the entire domain not only over the brighter snow surfaces as discussed latter. Therefore, explain in detail line 545 to 550. D)Detailed analysis and discussion should be done to explain why on doubling the dust concentration changes the sign/magnitude of dust induced perturbations on various variables nonlinearly (compared to that with initial dust). For example, in figure 14e, peak dust SDE in may end is ~1.5 Wm2 and the corresponding dust-induced reduction in SWE SWE is ~7% for Northern Rockies. This is very different from the situation in Figure 10 and 11. Dust SDE in Figure 11 is ~1W/m2 and the corresponding dust-induced reduction in SWE is ~ 2%. Please explain and discuss this nonlinearity.

2) A relative issue is this paper is very long which dilutes the main findings. I strongly suggest the author to significantly shrink the length of this paper by moving relatively minor parts/figures to supplementary and organization better to highlight the prime results.

3) Evaluation of snow cover duration should be included in the manuscript as authors report temporal shift in SWE as a main result.

4) Figure 10: The variability in runoff perturbation should be sum of perturbations in precipitation and that in SWE, But this is not the case in Northern Rockies. The precipitation increase and SWE decrease, both are maximum in June, but the runoff maximum is in May, why? This is not clear and need analysis and discussion.

5) In the text, the study period is mentioned as march through June, but in figure 7 it

is February through July. Why?. Also, why is the evaluation period different from the period averaged for results. It should be consistent.

6) I feel BCD is a misnomer and should be better described as LAP, a common term in literature for these light absorbing particles.

7) The authors discuss the differences in this study to previous modelling studies (Wu et al 2018 and Qian et al.,2009) over the same region in detail. One important difference between these 3 simulations is that they all are simulated at different spatial resolutions. The observed differences in the results related to surface elevation could also be due to the inherent variability in terrain height and thus snow depth and associated BCD-in-snow concentrations as also shown in a recent study by Sarangi et al.,2019, ACP (https://www.atmos-chem-phys.net/19/7105/2019/). This should be discussed in context.

8) What is the difference between ISWE and SDE in the manuscript, it seems to be same and used inter-changeably. Again, what is the definition and formula for calculating surface RE? We don't see good spatial correlation between surface RE and corresponding 2-m temperature in many figures? Why? Please define these terms and calculations clearly in methodology near Section 2.4.

9) Include tables like 3 and 4 for all the variables discussed in the manuscript.

10) Line 479» it should be aerosol

11) Line 560»it should be difference

---

## Author Comment (AC1) · 19 Jun 2020

Responses to reviewer RC1

We thank the reviewer for their helpful and insightful comments. We have done our best to address each concern.

The authors present a modeling study on the impact of black carbon (BC) and dust on the regional climate of the Rocky Mountains. Using WRF-Chem they examined the radiative impact of BC and dust in the atmosphere and the impact on the snow pack via

the modification of snow albedo and snow melting. They performed a series of simulations with all processes or with some processes eliminated. WRF-Chem simulations were limited to the period from February to July 2009 after spin-up simulations with WRF without chemistry. The simulations give important information on the contrasting radiative impacts of BC and dust in the atmosphere and the snow. For example, they confirm the larger radiative impact of BC compared to dust despite the orders of magnitude higher concentration of dust. The results also demonstrate the different regimes in four specific regions of the Rocky Mountains. The authors continue to discuss the potential impact of BC and dust on hydrological processes and specifically on the timing of the run-off. I have major concerns concerning this part of the manuscript, which in my opinion is less developed and less convincing. Therefore, I recommend major revisions before publication of the manuscript in ACP.

Reply: We thank the reviewer for the positive comments. The manuscript has undergone significant revisions. 4 Figures have been moved to the supplement. Of note, the acronym "BCD" has been changed to "light-absorbing particles" (LAPs) for better consistency with the literature. 5 appendices are now used to house more technical descriptions that weigh the paper down. Sec. 5.4 is now included along with Table 5 which highlights changes in meltout date.

Major comments (MaCs)

MaC1: In the simulations the CNT run only kicks in after 01/02/09. However, at this date approximately 60 % of the snow has on average already been deposited (Fig. 2c), but the BC and dust loading of this part of the snowpack is not known from the NOCHEM runs. How is this treated? Does the snowpack consist of a lower part of clean snow with layers including BC and dust on top? If yes, what is the impact on the simulations? How was this taken into account for calculated parameters (e.g. for the BC and dust in-snow burdens)?

Reply: This is an important detail. Snow at and beneath the surface was not initialized to a "clean" state. Originally, CNT was found to underpredict SWE substantially compared to measurements when initialized on 1 September instead of 1 February. Due to limited computational resources, a new modeling design was applied that saw the surface energy and hydrological fields from NOCHEM applied to the 1 February restart file, and in-snow BC and dust amounts were copied from the original WRF-Chem simulation to the new restart file where snow was present. This detail has now been included in the text: "We restart our WRF-Chem simulations on 1 February 2009 00:00 UTC using surface energy and hydrological fields from the NOCHEM restart file but in-snow LAP fields from the original WRF-Chem restart file." Specifically, LAP concentrations were used in all levels of the LSM where snow was present, and snow in the branch simulations was not initialized to a clean state.

MaC2: The authors claim in ch. 5.2 that since changes in simulated precipitation are at most 0.3 mm d-1 and SWE anomalies can be larger than 10 mm, the induced changes in SWE cannot be attributed to precipitation changes. I don't find this a convincing argument. Assuming that early in the winter season the solid precipitation increased by the given upper limit only for a period of a month and if all further processes remain unchanged, the resulting SWE for the rest of the winter season would increase by 9 mm. Therefore, the impact of precipitation changes in the simulations should be analyzed and discussed in more detail.

Reply: This point is well taken and understood, and this portion of the manuscript has been clarified to reflect this comment. ARI-induced precipitation modifications are generally less than 0.1 mm/d on average, not 0.3. However, it does appear that some correlation exists between the timing of ARI-induced precipitation and runoff anomalies. ARI-induced precipitation anomalies (now Fig. S4) correlate better with the ARI-induced runoff time series (Fig. 11c) than precipitation/runoff anomalies from SDEs (Figs. 9c, d, respectively). Furthermore, it seems as though snow changes are influencing runoff on a longer time scales than is precipitation. This is true for both SDEs and ARIs.

Addressing this point, the following changes have been made:

1. All statements suggesting that precipitation changes modulating runoff changes are negligible compared to SDEs have been removed from the manuscript, and the relevance of precipitation changes has been included for both ARI (Sec. 5.2.2) and SDEs (Sec. 5.1.2). In Sec. 5.2.2, the following paragraphs now read as follows:

"SWE (runoff) increases (decreases) from April onward due to LAP ARI across all four subregions prior to mid-May. These ARI-induced runoff changes are occurring in the presence of near-zero and nearly trendless precipitation (Fig. S4) and snowfall (not shown) anomalies. The simulation of these features suggests that the main driver of runoff changes, at least from April through mid-May, is depressed snowmelt from LAP ARI surface dimming. ARI-induced precipitation changes do impact runoff, however. For example, decreased precipitation from mid-May through 1 June (Fig. S4) correlates with decreased runoff during the same time period across Greater Idaho and the Northern Rockies (Fig. 11c). Following 1 June, runoff anomalies become less negative and even positive across the four subregions, a pattern opposite to that of LAP SDE (Fig. 9d). BC ARI tend to drive a majority of the runoff decreases prior to 1 June and promote increased runoff deeper into the summer. Dust ARI on the other hand has the opposite effect on runoff to that of BC ARI, increasing runoff through mid-May and decreasing runoff after 1 June across the Northern Rockies.

Comparatively, although SDE- and ARI-induced precipitation anomalies are of similar magnitude across the four subregions, the relative impact of LAP ARI-induced precipitation changes on runoff anomalies is larger than that of LAP SDE because the overall SWE changes associated with LAP SDE are larger. Larger snow (and subsequent runoff) changes occur due to LAP SDE, making the relative contribution of LAP SDE-induced precipitation changes to the total runoff changes smaller. Snowmelt and precipitation-specific runoff contributions were not output and thus cannot be explored further in this study."

In Sec. 5.2.1, the following paragraphs now read as follows:

"SDE-induced anomalies in SWE (Fig. 9b) and precipitation (Fig. 9c) change runoff by fractions of millimeters per day across the four subregions. Here, runoff is defined to be the sum of surface and underground runoff from the model output; runoff from glaciers and lakes is neglected. Driven primarily by BC SDEs, runoff is mostly increased through late June across all four subregions. Maximum simulated precipitation anomalies are generally less than 0.1 mm d-1 (Fig. 9c), while runoff anomalies are typically an order of magnitude larger (Fig. 10d). The largest increase in runoff occurs across the Northern Rockies (5.5 mm d-1, a 90% change from CNT), which is characterized by the largest reductions in SWE. During July, negative anomalies in runoff manifest, with the largest reductions simulated across the Northern Rockies (5.5 mm d-1, July mean ∼1%) and the Southern Rockies (4.5 mm d-1, July mean ∼2%). Smaller runoff reductions of ∼1 mm d-1 (2%) are simulated across Greater Idaho, while runoff increases of 1 to 2 mm d-1 (< 5%) are simulated across the Utah Mountains in phase with precipitation increases across this subregion (Fig. 10c). Although Qian et al. (2009) and Wu et al. (2018) emphasized results across basins, the dipole signature of runoff increases followed by runoff decreases is consistent with our results, despite primarily examining SDEs at higher elevations in this study. SDE-induced precipitation perturbations of greater than 0.1 mm d-1 are not simulated until mid-May, but runoff increases due to SDE are simulated beginning around 1 April. In the absence of a coherent trend in SDE-induced ice (not shown) or overall precipitation (Fig. 10c), we surmise that, at least initially, SDE-induced runoff anomalies are mainly driven by the enhanced melting of SWE and not SDE-induced precipitation changes. By mid-May, runoff increases across the Northern Rockies are relatively maximized, even as near-zero or slightly negative precipitation anomalies due to LAP SDE are simulated. There are however some correlations between the runoff time series and precipitation anomalies. For example, a local minimum in the runoff anomaly time series (Fig. 9d) is simulated around 1 June which correlates with negative precipitation anomalies of 0.3 mm d-1 across the Northern Rockies. In effect, this negative precipitation anomaly

[Figure]

is depressing the enhanced runoff signature induced by LAP SDE-induced snowmelt. During mid-June, precipitation increases in excess of 0.4 mm d-1 correlate with an increased positivity to the runoff anomaly time series (Fig. 9d) across Greater Idaho and the Northern Rockies."

2. Fig. S4 has been included showing ARI-induced perturbations to precipitation. MaC3: In general, the seasonal SWE average is in my opinion not an appropriate parameter since it includes the history of the precipitation. Solid precipitation early in the winter season has a larger impact on the average than later precipitation. The same is true for the simulation: if the SWE is modified early in the simulations the impact on the SWE average is larger than for later modifications. This leads than to confusing statements that the simulated SWE is larger than the observed SWE (e.g. ch 3.2), while Fig. 2c clearly show lower maximum SWE values in the NOCHEM and CNT runs. The positive bias probably stems only from the delayed snow melting in the simulations. Maybe, anomalies are better analyzed using SWE values at several specific dates? This could show the negative bias in spring and the positive bias in summer in the simulated SWE.

Reply: To clarify, SWE is underpredicted by CNT and NOCHEM compared to point SNOTEL observations, but SWE is overpredicted and underpredicted by CNT when compared to the spatial distribution from the UA product (Fig. 4; mostly overpredicted at higher elevations). Because CNT was not run for the full model year, it is impossible to take into account the SWE reductions due to LAP effects occurring prior to 1 February. CNT's overprediction of SWE at high elevations compared to UA occur where driving observations (e.g., SNOTEL) are scarce. Because the UA gridded product is driven by observations, the high modeled SWE bias at higher elevations may be artificial, as indicated in Broxton et al. (2016). More generally, CNT simulates less snow than NOCHEM, meaning that the model that includes aerosol effects (CNT) integrates a solution more dissimilar to SNOTEL observations than NOCHEM. This is due to the fact that the atmospheric and land surface parameterizations in NOCHEM, some of

which are empirically based, already partially account for these processes implicitly simply by virtue of the inclusion of SDE and ARI in the measurements for which the parameterizations were originally developed. Our goal here was not to show that CNT was closer to observations than NOCHEM but rather to discuss the physical changes in Rocky Mountain weather and hydrology due to SDE and ARI in high-resolution simulations, which has not been previously studied in this manner across this region. The differences between CNT and NOCHEM are vast and are beyond the scope of this study. As a sanity check, we wanted to ensure that the CNT results were comparable to a more commonly used counterpart without chemistry (NOCHEM), and we wanted to ensure that both simulations compared reasonably well with observations. Additional text comparing CNT and NOCHEM is now provided in Appendix A4. We also note that internal model variability may obfuscate more coherent agreements between NOCHEM and CNT, as well as lead to seemingly strange SWE anomalies, especially in light of the reviewer's point.

MaC4: Fig. 2c demonstrates further that the dynamics of the snow melting are not reproduced by the model independent if it includes BC and dust or not. Including BC and dust seems to shift the melt-out dates of the snow by a couple of days, but the simulated melt-out still appears to be delayed on average by more than 20 days compared to the observations. Moreover, observed melting rates are significantly higher than simulated melting rates. This should be discussed in more detail. This bias leads for example to large simulated impacts of BC and dust in the snow on temperature, SWE and run-off in July, for which the observations show no or rather little snow on the ground.

Reply: Thank you for this comment. Indeed, both CNT and NOCHEM deviate from SNOTEL observations. Both underpredict SWE and melt out snow too late (by $\sim$20 days). Unsurprisingly, CNT simulates less SWE than NOCHEM due to the explicit presence of BCD effects in said simulation, but NOCHEM simulates a superior SWE curve (Fig. 2c) than CNT compared to observations. While the root differences between

CNT and NOCHEM are beyond the scope of this paper (now mentioned in Appendix A4), as are WRF's/CLM's slow melt out deviations from observations, the ramifications of poorly simulated snow dynamics should at least be mentioned in the manuscript in context with potential weaknesses in observations. Appendix A4 reads:

"The goal of this study is to quantify the impacts of LAP SDE and ARI on WUS weather and hydrology. This aim does not align with examining root causes of differences between CNT and NOCHEM, and its scope does not necessarily focus on WRF's overall deficiencies in simulating seasonal snow dynamics. Nonetheless, we do note that significant technical differences exist between NOCHEM and CNT which lead to their different results.

First, upon grid-cell saturation, NOCHEM's number concentration of activated aerosols is prescribed in the microphysics scheme to be 250 cm-3, while CNT's is calculated online accounting for the local aerosol characteristics. This difference is most certainly leading to differences in the simulated snow yields through changes in the precipitation efficiency of clouds (not examined), with CNT simulating a smaller wet precipitation bias than NOCHEM compared to SNOTEL observations. An additional notable difference between CNT and NOCHEM is the coupling of chemical species' optical properties to the radiation code in CNT; this process is entirely neglected in NOCHEM and is also most certainly contributing to differences in solutions between the two results. More generally, WRF without chemistry (NOCHEM) has traditionally been developed to emulate the observed planet as closely as possible even though the model itself is free of explicitly simulated and physically based chemical processes, both in its atmospheric component and its land surface model. This study is an example of an instance where the inclusion of chemistry into the model (CNT) does not necessarily improve model performance. In fact, it appears that the presence of chemistry in CNT actually worsens our results compared to NOCHEM, as NOCHEM simulates SWE values closer to SNOTEL (Fig .2c) than CNT. Additionally, WRF (and other models) has traditionally showcased difficulties in simulating the evolution and timing of seasonal snow

dynamics (Caldwell et al., 2009; Wu et al., 2017), and our study does not attempt to explore why these deficiencies exist. Here, both simulations simulate a melt-out date ∼20 days later than is observed by SNOTEL. The differences between CNT and NOCHEM, as well as their deficiencies, should be kept in mind when interpreting the results of the study, and an evaluation of their differences is beyond the scope of this study."

Additionally, we have added Sec. 5.4 which quantifies the changes in meltout date. These results are summarized in newly added Table 5. Because meltout did not occur across 3/4 of our subregions, we present a "lag" time between CNT and their perturbation experiments. It was generally found that LAP effects accelerate meltout by ∼3-4 days.

MaC5: In the manuscript the hydrological impact is directly linked to surface run-off related to snow melting, without taking into account any detailed hydrological processes like groundwater storage or sub-surface transfer. This should be mentioned in the manuscript and potential impacts should be discussed. Moreover, since the dynamics and the timing of the snowpack melting in the simulations do appear to be biased (see above), it appears likely that the derived run-off is also strongly biased. How reliable are the conclusions concerning shifts in the timing of the run-off? A comparison with observed run-off data like for the atmospheric and snow data would be very helpful to support the conclusions in this part of the manuscript. In my opinion, related to this bias the simulations can at most give relative changes according to run off shifts in the runs with and without BC and dust. In my opinion, the presented shifts in run-off are not realistic and can in its current form not be used to inform local stakeholders. I recommend deleting from the manuscript all results and further parts describing and discussing the derived run-off.

Reply: While a complete water table analysis was not conducted in this study, the runoff results in this study reflect the changes in surface + subsurface runoff (mentioned in Sec. 5.1.2 as runoff deviations are presented). We did not output other variables such as runoff from glaciers, or groundwater storage/transfer that would have allowed us to

do a complete water budget analysis.

Regarding the reviewer's second point, the bias in simulated snow dynamics may indeed be biasing our results. However, intuitively, one might expect that a warming of the snow would accelerate snowmelt as winter transitions to summer, accelerating runoff. By late-spring and early summer, runoff rates would be depressed as a consequence of smaller snow amounts than baseline. This dipole runoff signature shows up in our sensitivity experiments and is consistent with the results of Qian et al. (2009, 2010) and Wu et al. (2018) focused on the western U.S. and indeed other regions (e.g., Rahimi et al. 2019). Moreover, these sensitivity experiments were run to explore if the findings of previous studies, which were conducted on comparatively coarse resolution grids, still held up at convective-permitting scales within a fully non-hydrostatic atmospheric model coupled with chemistry. Although the necessary output required to perform a complete water budget analysis is not available, the linkages between snowpack changes, precipitation changes, and runoff changes due to LAP effects can still be discussed in context to one another; these variables are fundamental to the local water budget across the intermountain west.

Regarding the reliability of these results, the changes in temperature, snow, precipitation, and runoff are comparable to results in previous studies (mentioned above). Hence, LAP-induced anomalies in these variables can be considered to be at least "physically plausible," if not "reliable", even if there are aspects of the overall meteorology that are simulated with inaccuracies (e.g., the snow dynamics). This will always be the case however in any form of numerical modeling framework. We believe however for these results to be truly informative for policymaking efforts, these experiments need to be conducted on multi-year time scales to develop a base climatology and smooth out internal model variability.

Finally, we examine how simulated temperature, precipitation, and snow compare to observations due to an abundance of high-resolution observational data products. We then include runoff in our analyses as an extension of our results, as (too our knowledge) runoff datasets are too coarse to capture fine-scale signatures across our domain. Because all products in this study were either point-source observations or characterized by grid spacings < 5 km, no evaluation of simulated runoff compared to observed runoff was performed; we were unable to find such high-resolution runoff datasets. For now, we keep simulated runoff in our analyses without validation.

Minor comments (MiCs)

MiC1: Concerning the impact of a modified snowpack on the hydrology of the western part of the US the authors refer in the introduction to Serreze et al., 1999 and Hamlet et al, 2007, which are both based on data from the last century. Adding studies on this subject based on more recent observations would be valuable for the readers.

Reply: The following citations have been added to reinforce the text based on more recent studies/observations:

Fyfe, J. C., Derksen, C., Mudryk, L., Flato, G. M., Santer, B. D., Swart, N. C., Molotch, N. P., Zhang, X., Wan, H., Arora, V. K., Scinocca, J. and Jiao, Y.: Large near-term projected snowpack loss over the western United States, Nat Commun, 8(1), 14996, doi:10.1038/ncomms14996, 2017.

Kapnick, S. and Hall, A.: Causes of recent changes in western North American snowpack, Clim Dyn, 38(9–10), 1885–1899, doi:10.1007/s00382-011-1089-y, 2012.

Mote, P. W., Li, S., Lettenmaier, D. P., Xiao, M. and Engel, R.: Dramatic declines in snowpack in the western US, npj Clim Atmos Sci, 1(1), 2, doi:10.1038/s41612-018-0012-1, 2018.

MiC2: It appears that the used emissions covered 2011, while the simulations covered the first half of 2009. It remains unclear for which year the boundary conditions are valid. Any specific conditions during any of the considered years? The potential impact should be briefly discussed.

Reply: This is a very good point. We have made sure to indicate which emissions data

[Figure]
* * *
Interactive
comment

are and are not simultaneous to our experimental period. Specifically, anthropogenic emissions are from 2011 inventories (non-simultaneous with our simulation period), but boundary condition and initial condition chemistry from MOZART-4, as well as fire emissions from FINN, are date-time specific to our experimental period. The following modifications to the text have been made in Sec. 2.2:

"Anthropogenic emissions from the Environmental Protection Agency's (EPA) 2011 National Emission's Inventory (EPA NEI-11; https://www.epa.gov/air-emissions-inventories/2011-national-emissions-inventory-nei-data) are used. These emissions contain location-specific point and area source emissions and are interpolated to a 4-km grid using the open-source software emiss_v04.F (ftp://aftp.fsl.noaa.gov); anthropogenic emissions from EPA NEI-11 are not simultaneous with our experimental time period. Simultaneous biomass burning emissions. . ."

Sec 2.3 has also been modified:

Reply: "Chemical boundary tendencies are updated every 6 hours beginning on 1 February 2009. MOZART-4 chemical input into WRF-Chem is date and time specific, but we note that in-domain anthropogenic emissions are averaged for the year 2011"

MiC3: It would be good to recall in ch. 2.1 how the introduction of BC and dust into the snowpack due to dry and wet deposition is treated in the SNICAR model and if and how BC and dust are preserved in the snow during melting.

Reply: The following discussion about SNICAR has been added as an appendix (A1):

"Simulated snow modification by the SNICAR model begins with LAP deposition flux (wet and dry) information calculated by the atmospheric chemistry module. As described in Flanner et al. (2012) and Zhao et al. (2014), dust (BC) mixes externally (internally and externally) with falling hydrometeors and is deposited on the snowpack.

Upon deposition, LAP is uniformly and immediately mixed throughout the layer. For BC, offline calculated Mie parameters (i.e., asymmetry parameter, SSA, extinction) valid for

effective radii of 0.1 mm are used from Chang and Charalampopoulos (1990). These values were used to derive snow absorption enhancement factors for a broad range of snow grain sizes. The mass absorption cross sections of BC are scaled by these factors which are found in a lookup table. For dust, optical properties in snowpack are derived from a combination of the Maxwell-Garnett mixing approximation and Mie theory. An assumed dust composition is used, and its size distribution is defined lognormally with a number median radius of 0.414 mm and a standard deviation of 2. Snow grains are treated by SNICAR as a collection of ice spheres with effective median number radii between 30-1500 mm. Mie parameters for snow are computed in one visible and four near-infrared bands offline. For the final radiative transfer calculations, BC, dust, and snow grains are treated as an external mixture by summing the extinction optical depths for each element, weighting the individual SSAs by the optical depths, and weighting the asymmetry parameters by the product of optical depths and the SSAs (Zhao et al., 2014). More information on the methods used in SNCAR can be found in Flanner et al. (2012). As the snowpack melts, meltwater scavenging of LAP is accounted for in SNICAR. Each layer in CLM4 has a threshold liquid capacity. Once this capacity is exceeded in a layer, the excess liquid is added to the liquid content of the layer beneath. The amount of scavenged LAP in this meltwater is proportional to this excess, the mass mixing ratio of LAP, and a scavenging factor (see Eq. 1; Zhao et al., 2014)." MiC4: In Fig. 2c it appears that the only significant difference between averaged SWE in CNT and NOCHEM occurs in the first half of March. Afterwards, the two curves seem to behave very similar with more or less constant differences. Is the impact of BC and dust in the snow on the simulated SWE only apparent in this short period? For example, the authors could show in Fig. 2c also the difference in SWE from CNT and NOCHEM to clarify this. I would actually expect that the impact is stronger during the melting phase than in March. If this is not the case, this should be discussed. Reply: It is impossible to pinpoint exactly what is driving the differences between CNT and NOCHEM here due to the fundamental differences between CNT and NOCHEM (addressed in MaC4). As mentioned, it was hoped that the CNT results

would be somewhat comparable to NOCHEM in terms of simulated temperature, precipitation, and snow properties, hence the motivation for running NOCHEM. NOCHEM was meant to serve not only as a starting point for the control (CNT) and perturbation WRF-Chem hydrological fields but also as another dataset to evaluate the performance of CNT. For examining the effects of LAP SDE and ARI on WUS weather and hydrology, we only use CNT and variations in CNT (perturbation experiments; noSDE, noARI, etc.); NOCHEM is not used in any analyses after Sec. 3.

Finally, it is indeed the case that LAP effects induce the largest changes in weather and runoff as the spring progresses. As indicated by time series in Figs. 9, 11, and 12 the largest perturbations to SWE (and other variables) are simulated from April through June (the melting season).

MiC5: The data shown in Fig. 2c cover a huge area. It would be useful to show the same curves also for the four selected regions, which exhibit in the simulations different snow dynamics as discussed later on the manuscript. Are there similar differences in observed and simulated SWE in the four specific regions?

Reply: Great idea. We have added Fig. S1 to the supplement. As can be seen, CNT simulates slightly less SWE than NOCHEM across all subregions. Furthermore, all CNT and NOCHEM melt out snow too late compared to SNOTEL, except across Greater Idaho. Greater Idaho sees the largest low bias in simulated SWE compared to SNOTEL and melts out snow ∼10 days later than is observed, while the Utah Mountains see simulated melt out occurring almost 30 days later than SNOTEL observations. Simulations do a fair job of reproducing the observed timing of maximized SWE in mid-April compared to SNOTEL regardless of subregion. Characterization of the snow melt out discrepancy is now presented in Sec. 3.1.

MiC6: The description of the impact of BC on snow metamorphism in lines 383ff appears rather superficial. The presence of absorbers in the snow has multiple impacts on the properties of the snow, which finally contribute to the radiative forcing. More

detailed descriptions of the processes can for example be found in Painter et al., 2007 and Flanner et al., 2007.

Reply: The snow-aerosol-albedo feedback enhancement by snow impurities was only skimmed in the intro. The paragraph in question has been modified to be more specific about what is happening regarding snow impurities and the enhancement of the snow-albedo feedback:

"The additional energy in the snowpack (Figs 7a, 7b, and S3) for a given time increases melting rates, leading to ice crystal growth of the underlying snow at the expense of liquid; larger ice crystals have a lower albedo than smaller ice crystals (Hadley and Kirchstetter, 2012). Increased heat content at the surface can warm the interfacing air via conduction, and this warming in turn melts more top snow, completing this feedback. Fig. 8j shows that mean snow grain radii are mostly enhanced by several microns across snow-covered regions from March through June. This enhancement in the snow-albedo feedback is explored in detail in Flanner et al. (2007) and Painter et al. (2007)."

References

Broxton, P. D., Dawson, N., and Zeng, X.: Linking snowfall and snow accumulation to generate spatial maps of SWE and snow depth, Earth Space Sci., 3, 246-256.

Caldwell, P., Chin, H.-N. S., Bader, D. C. and Bala, G.: Evaluation of a WRF dynamical downscaling simulation over California, Climatic Change, 95(3–4), 499–521, doi:10.1007/s10584-009-9583-5, 2009.

Chang, H. and Charalampopoulos, T. T.: Determination of the wavelength dependence of refractive indices of flame soot, P. Roy. Soc. Lond. A Mat., 430, 577–591, 1990.

Flanner, M. G., Liu, X., Zhou, C., Penner, J. E., and Jiao, C.: Enhanced solar energy absorption by internally-mixed black carbon in snow grains, Atmos. Chem. Phys., 12, 4699–4721, doi:10.5194/acp-12-4699-2012, 2012.

Fyfe, J. C., Derksen, C., Mudryk, L., Flato, G. M., Santer, B. D., Swart, N. C., Molotch, N. P., Zhang, X., Wan, H., Arora, V. K., Scinocca, J. and Jiao, Y.: Large near-term projected snowpack loss over the western United States, Nat Commun, 8(1), 14996, doi:10.1038/ncomms14996, 2017.

Kapnick, S. and Hall, A.: Causes of recent changes in western North American snowpack, Clim Dyn, 38(9–10), 1885–1899, doi:10.1007/s00382-011-1089-y, 2012.

Mote, P. W., Li, S., Lettenmaier, D. P., Xiao, M. and Engel, R.: Dramatic declines in snowpack in the western US, npj Clim Atmos Sci, 1(1), 2, doi:10.1038/s41612-018-0012-1, 2018.

Qian, Y., Gustafson, W. I., Leung, L. R. and Ghan, S. J.: Effects of soot-induced snow albedo change on snowpack and hydrological cycle in western United States based on Weather Research and Forecasting chemistry and regional climate simulations, Journal of Geophysical Research: Atmospheres, 114(D3), doi:10.1029/2008JD011039, 2009.

Qian, Y., Ghan, S. J. and Leung, L. R.: Downscaling hydroclimatic changes over the Western US based on CAM subgrid scheme and WRF regional climate simulations, International Journal of Climatology, 30(5), 675–693, doi:10.1002/joc.1928, 2010.

Rahimi, S., Liu, X., Wu, C., Lau, W. K., Brown, H., Wu, M. and Qian, Y.: Quantifying snow darkening and atmospheric radiative effects of black carbon and dust on the South Asian monsoon and hydrological cycle: experiments using variable-resolution CESM, Atmospheric Chemistry and Physics, 19(18), 12025–12049, doi:https://doi.org/10.5194/acp-19-12025-2019, 2019.

Wu, C., Liu, X., Lin, Z., Rhoades, A. M., Ullrich, P. A., Zarzycki, C. M., Lu, Z. and Rahimi‐Esfarjani, S. R.: Exploring a Variable-Resolution Approach for Simulating Regional Climate in the Rocky Mountain Region Using the VR-CESM, J. Geophys. Res., 122(20), 10,939-10,965, doi:10.1002/2017JD027008, 2017.

[Figure]

Wu, C., Liu, X., Lin, Z., Rahimi-Esfarjani, S. R. and Lu, Z.: Impacts of absorbing aerosol deposition on snowpack and hydrologic cycle in the Rocky Mountain region based on variable-resolution CESM (VR-CESM) simulations, Atmos. Chem. Phys., 18(2), 511–533, doi:10.5194/acp-18-511-2018, 2018.

Please also note the supplement to this comment:
https://www.atmos-chem-phys-discuss.net/acp-2019-998/acp-2019-998-AC1-supplement.pdf

―――――――――――――――――――――

---

## Author Comment (AC2) · 19 Jun 2020

Response to reviewer RC2

We thank the reviewer for their helpful and insightful comments. We have done our best to address each concern.

Based on model simulations, the authors examine the skill of high resolution WRFChem on the impact of snow albedo darkening and radiative forcing over western USA. They evaluate the model simulation with various observations. The authors also

discussed the radiative differences between BC and dust and intercompared two different pathways (snow direct radiative effect and atmosphere direct radiative effect) of aerosol effect on surface water budget. The spatial and temporal variation of radiative effects are also discussed. The experiments and results are interesting and suitable for publication in ACP after major revision to address following concerns. Reply: We thank the reviewer for the positive comments. The manuscript has undergone significant revisions. 4 Figures have been moved to the supplement, Of note, the acronym "BCD" has been changed to "light-absorbing particles" (LAPs) for better consistency with the literature. 5 appendices are now used to house more technical descriptions that weigh the paper down, and other text has been moved to the supplement. Sec. 5.4 is now included along with Table 5 which highlights changes in meltout date. Major comments (MaCs) MaC1: More discussion on role of dust in the manuscript is needed along with clear explanation and analysis. A) why is dust induced SWE positive over Northern Rockies in Figure 8f,which is inconsistent with the fact that dust-ISRE is positive (Figure 7d) and dust-induced change in albedo is negative (Figure 8i). B) More aerosols always have a negative radiative effect at surface as it either scatters or absorbs incoming radiation at surface. Why is dust surface RE in Figure 12i positive? Although, the authors have tried to explain this by stating the differences in aerosol microphysics, I feel it is not clear. C) Also, this is found over the entire domain not only over the brighter snow surfaces as discussed latter. Therefore, explain in detail line 545 to 550. D)Detailed analysis and discussion should be done to explain why on doubling the dust concentration changes the sign/magnitude of dust induced perturbations on various variables nonlinearly (compared to that with initial dust). For example, in figure 14e, peak dust SDE in may end is ∼1.5 Wm2 and the corresponding dust-induced reduction in SWE SWE is ∼7% for Northern Rockies. This is very different from the situation in Figure 10 and 11. Dust SDE in Figure 11 is ∼1W/m2 and the corresponding dust-induced reduction in SWE is ∼ 2%. Please explain and discuss this nonlinearity.

Reply: We respond to this comment in segments below.

Why is dust induced SWE positive over Northern Rockies in Figure 8f, which is inconsistent with the fact that dust-ISRE is positive (Figure 7d) and dust-induced change in albedo is negative (Figure 8i).

Reply: We believe this peculiarity was the result of internal model variability. Simulations integrate independently even though they are initialized to the same file. Small differences in simulated physics (i.e. certain physical effects were disabled in certain experiments) led to small changes in storm timing/location in the different experiments. At some points characterized by positive SWE anomalies, even a single storm difference, specifically a difference that saw less snowfall in CNT, was enough to yield a March-June mean anomaly that was positive. Regionally averaged, it is clear that BC and dust SDEs contribute to SWE reductions (see Fig. 9b). Furthermore, dust SDE (and ARI) anomalies were assumed to be the linear difference between BC+dust SDE (ARI) and BC SDE (ARI) effects. This linearity assumption may have partially contributed to the positive SDE anomalies. To affirm our suspicious that internal model variability was driving strange SWE anomalies, we examined correlations between SWE, temperature, precipitation, cloud, and circulation anomalies with terrain height; they were unrevealing, and no coherent signature was found. It is believed that these positive SWE anomalies would most likely be uncommon if we conducted our experiments over many years and averaged over climate-relevant time scales. As such, we have added the following to the Sec. 5.1.1:

"We note that there are areas where LAP SDE leads to increased SWE amounts across a small fraction of gridcells (Fig. 8d). We believe this to be the result of internal model variability rather than a physical manifestation (Bassett et al. 2020). Examination of several grid points where the March-through-June mean SWE anomalies were positive revealed that fine-scale storm location and intensity differences between, for instance, CNT and noBCSDE were leading to positive SWE anomalies (not shown). We expect these positive SDE-induced SWE anomalies to be more uncommon if averaged over climate-relevant time scales. As will be shown in the next section, SDE SWE anomalies

are negative when averaged regionally."

Internal model variability must also be considered when interpreting ARI-induced anomalies. As such, the following sentence was added to Sec. 5.2.1, P2: "As with SDEs, internal model variability may be responsible for unintuitive SWE anomalies, but this issue was not examined in this study due to limited computational resources."

More aerosols always have a negative radiative effect at surface as it either scatters or absorbs incoming radiation at surface. Why is dust surface RE in Figure 12i positive? Although, the authors have tried to explain this by stating the differences in aerosol microphysics, I feel it is not clear.

Reply: The surface radiative effects (REs) for BC+dust, BC, and dust are shown in Figs. 10g, h, and I, respectively. RE values are the sum of the shortwave and longwave REs. The key difference between BC and dust is that dust aerosols can downwell longwave energy. For dust aerosols of the right size and number concentration (Tegen and Lecis, 2012), this downwelled longwave energy can compensate for and even exceed solar dimming, yielding positive surface RE values (Fig. 10i). BC on the other hand is not a good attenuator of longwave energy because of its relatively small sizes. BC is however a very effective scatterer/absorber of incoming solar energy. Thus, BC dims the surface (Fig. 10h), yielding a very different surface RE compared to dust. BC-induced dimming dominates over dust-induced longwave warming, yielding a negative surface RE across the domain (Fig. 10g). The following discussion has been added to Sec 5.2.1 P3: "Specifically, the key difference between BC and dust is that dust aerosols can downwell several W m-2 of longwave energy. Depending on the dust size and number concentration, this downwelled longwave energy can dominate over the solar dimming, yielding positive surface RE values (Fig. 10i; Tegen and Lecis (2012)). BC meanwhile is not an effective attenuator of terrestrial energy because of its relatively small sizes. BC is however a very effective scatterer/absorber of incoming solar energy. Thus, BC dims the surface (Fig. 10h), yielding a very different surface RE compared to dust (Fig. 10i). BC-induced dimming exceeds the dust-induced longwave

RE, yielding a negative surface RE across the domain (Fig. 10g)."

Also, this is found over the entire domain not only over the brighter snow surfaces as discussed latter. Therefore, explain in detail line 545 to 550.

Reply: These lines provide an explanation of why the negative SWE anomalies from dust ARI tend to be maximized atop high-albedo surfaces of the Northern Rockies, especially since BC dimming dominates the LAP surface RE. Downwelled terrestrial radiation contributes positively to surface REs everywhere (Fig. 10i). Meanwhile, BC dims everywhere, with dimming decreased across high-albedo surfaces (Fig. 10h) by Eq. (3). Depressed solar dimming by BC over high-albedo surfaces coupled with ever-present downwelled terrestrial radiation by dust leads to a "less negative" total (BC+dust) RE at the surface and subsequently increased snow melting regionally averaged. Ultimately, the microphysical differences between the two aerosol types dictate in what wavebands they attenuate. The examination of these aerosols' microphysical nature is beyond the scope of this manuscript. Clarification to these lines are made: "For atmospheric dust (and BC) particles residing over the high-albedo surface of the Northern Rockies, this means that there will be a higher chance of shortwave absorption at the surface through a larger $S\_total^↓$. Together with dust longwave warming (Figs. 10i and 11d), dust ARI contribute to snowpack reductions across the Northern Rockies. SWE reductions are most prominent across the Northern Rockies subregion where smaller, more scattering dust particles are present. The physical process described here is similar to that noted in Stone et al. (2008) who examined the atmospheric REs of wildfire smoke across northern Alaska's high albedo surface."

Detailed analysis and discussion should be done to explain why on doubling the dust concentration changes the sign/magnitude of dust induced perturbations on various variables nonlinearly (compared to that with initial dust). For example, in figure 14e, peak dust SDE in may end is $\sim$1.5 Wm2 and the corresponding dust-induced reduction in SWE SWE is $\sim$7% for Northern Rockies. This is very different from the situation in Figure 10 and 11. Dust SDE in Figure 11 is $\sim$1W/m2 and the corresponding dustinduced reduction in SWE is ∼2%. Please explain and discuss this nonlinearity.

Reply: Unfortunately, we cannot estimate the efficacy of dust SDE SWE reductions without further experiments being conducted. Four specific factors must be considered when evaluating the effects of increased dust emissions (Fig. 13 and Sec. 6). 1. The original dust ARI (SDEs) are considered to be linear differences of the noBCSDE (noBCARI) from the noSDE (noARI) experiment. However, DTF=2-CNT anomalies represent linear+nolinear effects of increased dust emissions, 2. We cannot explicitly determine whether SDE or ARI enhancement associated with increased dust emissions is driving DTF=2-CNT anomalies; undoubtedly both dust SDE and dust ARI increase in DTF=2, 3. Internal model variability effects, and 4. Increased dust emissions can impact cloud properties (indirect effects), which are not a focus of this study. We acknowledge the well-documented nonlinearity between snow-albedo and the mass concentration of impurities (e.g. Hadley and Kirchstetter 2012; Flanner et al. 2007; Painter et al. 2009; Wiscombe and Warren, 1980)

March-through-June cloud fraction differences between DTF=2 and CNT were found to be < 2% and generally positive, so (4) is discounted as a possibility for increasing snowmelt in DTF=2. (3) is probably not the issue here, as a regional average somewhat smooths noise associated with internal model variability, hence the prominence of negative SWE anomalies in Fig.13e across the Northern Rockies. The in-snow dust as a percentage is increased the most in DTF=2 relative to CNT across the Northern Rockies compared to any other subregion due to the fact that CNT simulates relatively low top-snow dust amounts of 2-4 mg m-2 (Fig. 6b). These amounts are increased by 50-80% in DTF=2 by small dust particles emitted to the southwest. Clearly, the ratio of the top snow dust amount in DTF=2 to CNT is maximized across the Northern Rockies, even as the increase in ISRE is maximized across the Utah Mountains. This exemplifies the nonlinear relationship between snow impurity amount, ISRE, and SWE reductions. It is thus believed that because of increased dust ARI-induced warming in DTF=2 (see Sec. 5.5, Eq. 3, Fig. 13b), in addition to the larger fractional change in

snow impurities (and thus a stronger dust SDE), larger SWE reductions are simulated in DTF=2 than CNT across the Northern Rockies compared to the Utah Mountains. Of course, the assumption of linearity (1) in estimating the original dust SDE/ARI complicates this comparison, as DTF=2-CNT anomalies include linear and nonlinear effects. In any case, the main point of conducting the DTF=2 experiment was to assess the impacts of the simulated low dust bias, not to assess the effects of our linear assumption in assessing dust effects. We cannot completely address this comment without conducting further experiments. The following paragraph has been added to Sec. 6:

"While it can be seen that increased dust emissions have consequences on simulated meteorology, it cannot be determined whether a majority of changes in meteorological variables are due to enhancements in dust SDE or dust ARI without conducting further experiments. We did identify small increases in cloud amounts (by less than 2%; not shown). In-snow dust burdens, as a percentage, were increased the most across the northern subregions, although ISRE perturbations in DTF=2 were smaller compared to the southern subregions (Fig. 13a). However, perturbations to the surface RE were generally positive across high-elevation areas of the northern subregions, especially the Northern Rockies (Fig. 13b). Evaluating enhancements in dust SDE in the DTF=2 experiment is complicated by the nonlinear relationship between snow impurity amount and radiation absorption (Flanner et al. 2007, 2012; Painter et al., 2007; Hadley and Kirchstetter 2012; Wiscombe and Warren, 1980). DTF=2 enhancements of dust effects over CNT comprise linear and nonlinear ARI and SDE, whereas earlier computation of dust ARI and SDE were subject to a linearity assumption, further complicating the comparison of DTF=2-CNT anomalies with previously computed dust anomalies (Secs. 5.1, 5.2). We emphasize the limitations of our assumptions in quantifying dust effects, and call for further studies of dust SDE and ARI across this region.

MaC2: A relative issue is this paper is very long which dilutes the main findings. I strongly suggest the author to significantly shrink the length of this paper by moving relatively minor parts/figures to supplementary and organization better to highlight the

prime results.

Reply: Following revisions, the paper has a higher word count, but it has been significantly reorganized to present a less diluted product. Four figures have been added to the supplement. Also, 1 supplemental subsection and 5 appendices have been created to highlight less important details of the manuscript.

MaC3: Evaluation of snow cover duration should be included in the manuscript as authors report temporal shift in SWE as a main result

Reply: As can be seen if Fig. S1, CNT (and NOCHEM) melt out snow far too late compared to SNOTEL, especially across the Western WY, Utah, and Colorado (note these are regional averages of SNOTEL point observations, not our defined subregions). For our defined subregions (i.e. Greater Idaho, Northern Rockies, Utah Mountains, and Southern Rockies), melt out dates are also overpredicted by CNT be multiple weeks, which is consistent with the CNT/SNOTEL bias. In fact, meltout (SWE = 0 mm) is simulated only in Greater Idaho in CNT; all other subregions have simulated SWE on 1 August. Because our simulations were not run long enough to explicitly capture meltout on regional scales, we did not initially include these results in the manuscript.

With this is mind, we did evaluate snowmelt timing by comparing 1 August SWE in CNT with other experiments across our four subregions. For instance, if SWE = 15 mm in CNT and 18 mm in noBCD, we would find the day in noBCD that had a SWE value of 15 mm and compute the lag time. We found this lag time and used it as a proxy for meltout shifts for subregions in which we did not actually simulate melt out. These results are presented in Table 5. Section 5.4, "Changes in meltout timing", has been added to discuss meltout changes.

MaC4: Figure 10: The variability in runoff perturbation should be sum of perturbations in precipitation and that in SWE, But this is not the case in Northern Rockies. The precipitation increase and SWE decrease, both are maximum in June, but the runoff maximum is in May, why? This is not clear and need analysis and discussion.

[Figure]

Reply: Correct, runoff comes from both precipitation and snowmelt, but we only consider the total runoff, and we did not output runoff from snowmelt and precipitation individually. It can be seen in Fig. 9c that precipitation anomalies are negative from late May through early June across the Northern Rockies. Meanwhile, runoff from accelerated snowmelt is presumably positive as runoff due to precipitation is reduced. We thus see a local minimum in runoff from late May into early June that coincides nicely with the negative precipitation anomalies. Runoff is still positive due to enhanced snowmelt by LAP effects.

MaC5: In the text, the study period is mentioned as march through June, but in figure 7 it is February through July. Why? Also, why is the evaluation period different from the period averaged for results. It should be consistent.

Reply: Nice catch! We chose to emphasize the March-through-June time frame in our spatial distribution figures highlighting LAP effects/properties because 1. Aerosol burdens are either increasing or are maximized during this time period, 2. The solar elevation is increasing, and 3. Snowpack is maximized. Thus, aerosol effects emphasized in this study should be maximized during this time period. We also wanted to increase the signal-to-noise ratio. As such, Figs. 7 and S2 showing the spatial distribution of ISRE/RE and aerosol burdens, respectively, have been changed to a March-through-June average, consistent with other aerosol-related figures. Otherwise, the meteorological figures/metrics remain a February-through-July average; their biases do not change significantly when averaged from March through June, so we keep the averages in Sec. 3.

Mac6: I feel BCD is a misnomer and should be better described as LAP, a common term in literature for these light absorbing particles.

Reply: Done

Mac7: The authors discuss the differences in this study to previous modelling studies (Wu et al 2018 and Qian et al.,2009) over the same region in detail. One important

difference between these 3 simulations is that they all are simulated at different spatial resolutions. The observed differences in the results related to surface elevation could also be due to the inherent variability in terrain height and thus snow depth and associated BCD-in-snow concentrations as also shown in a recent study by Sarangi et al.,2019, ACP (https://www.atmos-chem-phys.net/19/7105/2019/). This should be discussed in context.

Reply: This is a very significant paper and reinforces the motivation for this study. As such, the paper has been cited three times in the introduction.

MaC8: What is the difference between ISRE and SDE in the manuscript, it seems to be same and used inter-changeably. Again, what is the definition and formula for calculating surface RE? We don't see good spatial correlation between surface RE and corresponding 2-m temperature in many figures? Why? Please define these terms and calculations clearly in methodology near Section 2.4.

Reply: ISRE is the excess energy absorbed by snow due to impurities [W m-2] and is only physical where there is simulated snow. The SDE describes a physical process which begins with the deposition of LAPs on snow. This slightly darker snow absorbs more incoming solar energy, increases snowmelt, etc. The surface RE is computed using the methods in Ghan et a. (2012) across all grid cells. The calculation of REs is briefly mentioned in Appendix A5 and involves two steps. First, all aerosol effects are accounted for in the radiative calculations for upwelled and downwelled energy. Second, the same section of the code is called again, but with specific aerosols' radiative properties disabled. By subtracting the former with the latter, REs can be computed for different aerosol species. As for the poor correlation between the surface RE and 2-m temperature (T2), the surface energy surplus can be converted to sensible heat, latent heat or it can be conducted away to the overlying air or the snow. Since we emphasize REs over snow covered regions, this excess heat is conducted to the snow by absorbing LAPs. The snow melts isothermally under a phase change and we therefore do not see a high correlation between positive T2 and larger REs, especially across the

highest elevations.

MaC9: Include tables like 3 and 4 for all the variables discussed in the manuscript.

Reply: We emphasize the time series of other variables in Fig. 9a-d, 11a-d, 12a-c, S3, and S4. We can convert to a table, but we believe this to be a better way to visualize LAP-induced perturbations to WUS weather.

MaC10: Line 479Âż it should be aerosol

Reply: Fixed

MaC11: Line 560Âżit should be difference

Reply: Fixed

References

Bassett, R., P. J. Young, G. S. Blair, F. Samreen, and W. Simm. "A Large Ensemble Approach to Quantifying Internal Model Variability Within the WRF Numerical Model." Journal of Geophysical Research: Atmospheres 125, no. 7 (2020): e2019JD031286. https://doi.org/10.1029/2019JD031286.

Flanner, M. G., Zender, C. S., Randerson, J. T. and Rasch, P. J.: Present-day climate forcing and response from black carbon in snow, J. Geophys. Res., 112(D11), D11202, doi:10.1029/2006JD008003, 2007.

Flanner, M. G., Liu, X., Zhou, C., Penner, J. E., and Jiao, C.: Enhanced solar energy absorption by internally-mixed black carbon in snow grains, Atmos. Chem. Phys., 12, 4699–4721, doi:10.5194/acp-12-4699-2012, 2012. Painter, T. H., Barrett, A. P., Landry, C. C., Neff, J. C., Cassidy, M. P., Lawrence, C. R., McBride, K. E., and Farmer, G. L.: Impact of disturbed desert soils on duration of mountain snow cover, Geophys. Res. Lett., 34, L12502, https://doi.org/10.1029/2007GL030284, 2007. Wiscombe, W. J. and Warren, S. G.: A model for the spectral albedo of snow, I: Pure snow, J. Atmos. Sci., 37, 2712–2733, 1980.

[Figure]

Please also note the supplement to this comment: https://www.atmos-chem-phys-discuss.net/acp-2019-998/acp-2019-998-AC2-supplement.pdf
* * *

---

## Author Response (AR2)

**Revisions – round 2**

**Reviewer 1**

In my opinion the authors did a very good job to address the major concerns expressed in my initial report. Including the appendices is a very good approach making the manuscript easier to read. Some appendices may even be moved to the supplement (e.g. A1 or A5) to reduce the length of the manuscript? The authors may further consider the suggestions for technical corrections below.

*Reply*: We once again thank the reviewer for their helpful comments.

Rockies should be replaced by Rocky Mountains throughout the manuscript.

*Reply*: The word "Rockies" is used in the names of two arbitrarily defined subregions used in this study: The Northern Rockies and the Southern Rockies. We use these names because they are relatively shorter than "Northern Rocky Mountains" and "Southern Rocky Mountains," and they are colloquially referred to as the "Rockies" We leave these names alone for now, but we have changed "Rockies" to "Rocky Mountains" in the manuscript where these subregions are not being referred to per this comment.

Chapter 2.5 Observational data: It is difficult to see in Fig. 1, but I assume that most of the IMPROVE observations are from relatively low elevation sites? Maybe the range of elevation of the used IMPROVE sites could be mentioned in the last paragraph of this chapter. This may be important since the bias between observed and simulated BC and dust may be higher at higher elevations.

*Reply*: Correct, and this is a very good point. Sites are typically at higher elevations; the mean elevation of our IMPROVE sites is right around 2200 m, ranging from 1195 m to 3413 m. The following has been added to P2 of Sec. 2.5: "These sites are at relatively high elevations (mean elevation 2,221 m) with the lowest station located at 1,195 m and the highest at 3,413 m."

L. 599: "…[dust] is still underpredicted by 43%."

*Reply*: Corrected.

Some sentences are somewhat awkward and should be revised, e.g. line 558 "Larger, more dust aerosols dim sunlight, effectuating a negative surface RE.", line 554 "Only in Greater Idaho is CNT able to explicitly meltout snow;… ", line 565 "…move up the meltout date by 4 days.", line 566ff "Painter et al. (2007) simulated a dust SDE that melted snowpack completely more than 20 days earlier than a simulation without dust-snow-albedo effects, a result of much larger magnitude than ours.", line 588 "However, by (3), it is clear that..", or line 662 "… dust ARI could actually incite a positive surface RE."

*Reply*: Corrected. The first referenced sentence was missing a word. In the second, "meltout" should be two words and has been corrected. In the third, we have changed "up" to "forward," as we are trying to indicate that the meltout date is occurring 4 days earlier due to LAP effects. The fourth referenced example now reads, "Painter et al. (2007) simulated a dust SDE that accelerated snowpack meltout by more than 20 days compared to a simulation without dust-snow-albedo effects, a result of much larger magnitude than ours." The fifth referenced clause now reads, "By (3), it is clear that when $f_a$ is negligible…" The final clause now reads, "…dust ARI could incite a positive or a positive surface RE…"

Typo in line 563: "…between SDE and ARI shifts"

*Reply*: Rewarded to: "…and we note that AIR and SDE-induced anomalies in meltout do not add linearly…"

Caption of Figure S1: Use the same denominations for the four sub-regions as in the rest of the manuscript and the supplement.

*Reply*: Corrected

**Reviewer 2**

The manuscript has improved a lot in terms of organisation, clarity in discussions and presentation of the results. The authors have addressed my comments and included discussions towards explaining the differences seen due to interplay between radiative differences of BC and dust over the snow surface.
I recommend publication of this work with minor comments.

*Reply*: We thank the reviewer for their helpful comments

1) I found that the numbering of Figures is wrong in the submitted version.

*Reply*: We have ensured that all figures are numbered and referenced properly.

2) Figure 9-13, show spatial mean values but do not have a standard deviation. Could be included for details.

*Reply*: With nearly 10 time series per subpanel, adding SD values makes the figures overly noisy and distracts from the main message. We can add the SD values if absolutely necessary though.